# Error Controlled Feature Selection for Ultrahigh Dimensional and Highly Correlated Feature Space Using Deep Learning

## Abstract

Deep learning has been at the center of analytics in recent years due to its impressive empirical success in analyzing complex data objects. Despite this success, most existing tools behave like black-box machines, thus the increasing interest in interpretable, reliable, and robust deep learning models applicable to a broad class of applications. Feature-selected deep learning has emerged as a promising tool in this realm. However, the recent developments do not accommodate ultra-high dimensional and highly correlated features or high noise levels. In this article, we propose a novel screening and cleaning method with the aid of deep learning for a data-adaptive multi-resolutional discovery of highly correlated predictors with a controlled error rate. Extensive empirical evaluations over a wide range of simulated scenarios and several real datasets demonstrate the effectiveness of the proposed method in achieving high power while keeping the false discovery rate at a minimum.

## 1 Introduction

In modern applications (e.g., genetics and imaging studies), the investigator is often interested in uncovering the true pattern between a quantitiave response and a large number of features. The key working assumption, oftentimes, is that there is an underlying sparsity pattern buried in the high dimensional data structure. Detecting the essential features plays a crucial role in facilitating scientific investigations by offering improved interpretability and explainability, coupled with reduced computational cost and less memory usage. Under the linear model (LM) framework, this problem has been extensively studied over the past few decades, stemming from the popular algorithms such as Lasso, Elastic net, SCAD, and MCP. A detailed review of this literature can be found elsewhere (e.g., Fan & Lv (2010)) and thus is omitted here. However, regardless of their ubiquitous applications, the LM has limited applications, especially when the underlying mechanism is highly nonlinear, with potential interaction effects. This is particularly critical given that the true underlying relationship is unknown to the analysts in real applications. The Artificial Neural Network (ANN) models that relax the linearity assumption are well known for efficiently approximating complicated functions. For example, from an information-theoretic viewpoint, Elbrächter et al. (2021) established that Deep Neural Networks (DNN) provide an optimal approximation of a nonlinear function, covering a wide range of functional classes used in signal processing. This property has promoted the use of Deep Learning (DL) models for feature selection, an approach that has generated much research interest over the past few years. However, the black-box nature and lack of explainability in DL models pose a major drawback, raising concerns about their applicability in real-world decision-making, as highlighted by Rudin (2019). Hence, it is crucial to exercise caution when employing DL models in practical applications.

**Error controlled feature selection:** Employing only the relevant predictors to construct a predictive model is the right step toward explainable machine learning. Moreover, recent studies have shown that the feature importance in DL-based algorithms can demonstrate substantial variability in the presence of small input perturbations or a lower signal-to-noise ratio Ghorbani et al. (2019). To address this challenge, our approach aims to achieve reproducible nonlinear error-controlled variable selection using DL models. In this regard, we employ the False Discovery Rate (FDR), originally proposed by Benjamini & Hochberg (1995). FDR is well-suited for large-scale multiple testing problems, offering a less conservative and more

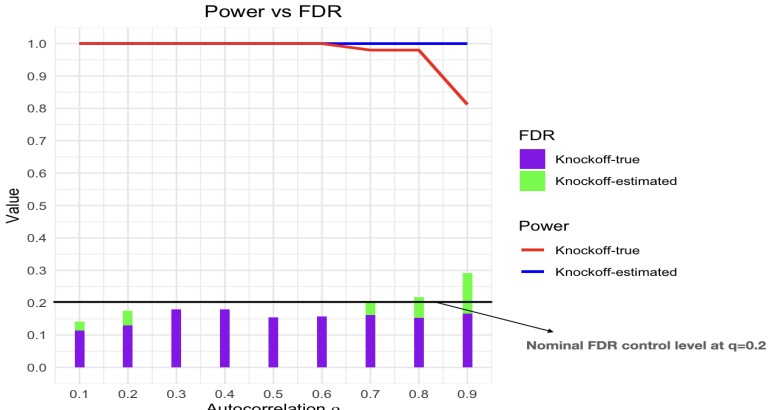

Figure 1: **Examining the Impact of Multicollinearity on Knockoffs - A demonstration through a simplistic simulation setting** We generate $n = 400$ independent and identically distributed samples ($y \in \mathcal{R}, X \in \mathcal{R}^{100}$), where the outcome $y$ is generated by the linear model: $y = 0.5(X_{20} + X_{40} + X_{60} + X_{80} + X_{100}) + \epsilon$, with $\epsilon \sim N(0, 20)$. The features $X$ follow a multivariate normal distribution $N_{100}(0, \Sigma)$, where $(\Sigma)_{ij} = \rho^{|i-j|}$ represents the autocorrelation strength. We vary the autocorrelation strength $\rho$ from 0.1 to 0.95. Two approaches for Model-X knockoff generation are implemented on 500 Monte Carlo replication of the data: (1) Model-X estimated, using the method proposed in Candès et al. (2016) to generate knockoffs based on the estimated feature distribution, and (2) Model-X True, generating knockoffs from the true distribution of $X$. The FDR (false discovery rate) is depicted by vertical bars along the left y-axis, while the associated power is represented by horizontal lines by the right y-axis. The black horizontal line indicates the nominal FDR control level at $q = 0.20$.

powerful alternative compared to traditional methods such as the Family Wise Error rate (FWER). To formally define the FDR, we consider the random variable, $FDP$, representing the False Discovery Proportion: $FDP = \frac{e_0}{N_+ \wedge 1}$, where $e_0 =$ number of falsely selected variables, $N_+ =$ number of total discoveries. Then, FDR is defined as $FDR = E(FDP)$. The estimation of this expectation, considering the unknown data-generating distribution, presents a distinct challenge in the context of model-free variable selection. To tackle this issue, various approaches have been proposed in the literature. For instance, multiple testing literature suggests using a p-value-based approach as a criterion for feature importance; refer to Tansey et al. (2018); Xia et al. (2017); Li & Barber (2019); Lei & Fithian (2018) for an overview. However, generating interpretable p-values remains an unresolved research issue in the context of DL models. To circumvent this limitation, the knockoff framework has been proposed by Candès et al. (2016). Essentially, this is a model-free variable selection algorithm with provable FDR control, assuming one has prior knowledge of the predictors' distribution. Lu et al. (2018) further proposed the *DeepPINK* algorithm by integrating the knockoff framework with the DL architecture for improved explainability of the DL models. Nevertheless, in real-world applications, it is often unrealistic to assume full knowledge of the distribution of the high-dimensional feature vector, which is a fundamental requirement for the knockoff approach. Consequently, the predictor's distribution needs to be estimated in a real-world application to generate the knockoff variables, which adds another layer of uncertainty to the analysis. Recently Barber et al. (2020) showed that the knockoff framework can lead to inflation in false discoveries, which is proportional to the error incurred in estimating the predictor's distribution. This problem is exacerbated by highly correlated features. Figure 1 provides an empirical illustration of Barber et al. (2020), showcasing how the model-X knockoff approach Candès et al. (2016) fails to effectively control the FDR under high multicollinearity. Figure 1 also demonstrates that the problem lies in accurately estimating the distribution of the features, as knockoffs generated from the true feature distribution successfully control the FDR. However, relying on the true distribution is often unrealistic in practical settings where the feature distribution is typically unknown. Consequently, any knockoff generation method that imposes distributional assumptions on the features, such as (Candès et al., 2016; Sesia et al.,

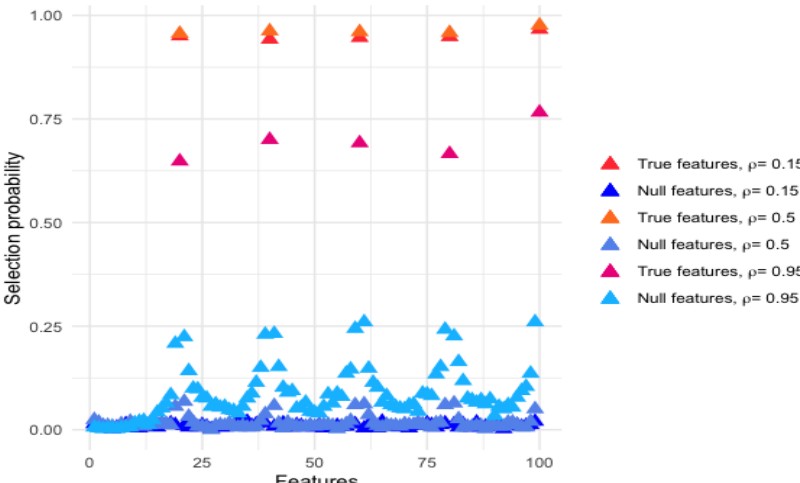

Figure 2: **How multicollinearity affects feature selection**. With the same simulation setting in the linear model in Figure 1, we implement the Lasso algorithm for 500 Monte Carlo replication of the data, and the y-axis shows the proportion of time each feature is selected out of 500 replications. For a higher autocorrelation $\rho$, the selection probability of the true features was significantly reduced whereas the null features that are associated with true features got selected more frequently.

2018), may prove inefficient in a general setting. To address these challenges associated with model-based knockoff methods, recent advancements have introduced flexible knockoff generation algorithms based on deep learning (DL), such as (Liu & Zheng, 2019; Jordon et al., 2019; Romano et al., 2020). however they are trained in a typical big-$n$-small-$p$ setting, and their performance in ultra-high dimensional settings with limited sample size remains unclear. We next discuss in detail the multicollinearity issue in the context of multi-resolutional feature selection.

**Addressing uncertainty in highly correlated feature space:** In contemporary high-dimensional datasets often found in genetics and imaging studies, the presence of extreme multicollinearity is a significant challenge in addition to ultra-high dimensionality. This phenomenon manifests as a complex intercorrelation among the predictors, often with pairwise sample correlations surpassing 0.99. In a simplistic linear model setting, Figure 2 shows how increased autocorrelation typically reduces the selection probability of the true non-null features. Due to the near-indistinguishability of extremely correlated features, it is unrealistic to assert that a specific feature within a cluster is independently associated with the outcome in a regression framework. Therefore, it is more pragmatic to address the uncertainty by pursuing group-level variable selection, asserting that at least one variable within a densely correlated group is significant for the outcome. As such, by '*true discovery*' one refers to the selection of clusters that can serve as good proxies for at least one element in the true index set of important features $S_0$. This approach has been discussed in the context of statistical genetics. For example, Brzyski et al. (2017)eloquently argued that the discovery of a specific genomic region is considered more informative than the discovery of individual variants within that region. Although conceptually appealing, a complication of this approach is that the notion of FDR in its original formulation becomes non-trivial. For this reason, following Siegmund et al. (2011), we adopt the cluster version of the FDR as the *expected value of the "proportion of clusters that are falsely declared among all declared clusters"*. We denote this as *cFDR* henceforth. There are other methods for handling extreme multicollinearity, which have for the most part appeared in the hierarchical testing literature. Few examples include *CAVIAR* (Hormozdiari et al., 2014), *SUSIE* (Wang et al., 2020), and *KnockoffZoom* (Sesia et al., 2019). While the knockoff-based procedures have the limitation of generating knockoffs from an unknown distribution with a very small sample size, other methods lack applicability in non-linear-nonparametric setups as they typically depend on p-values.

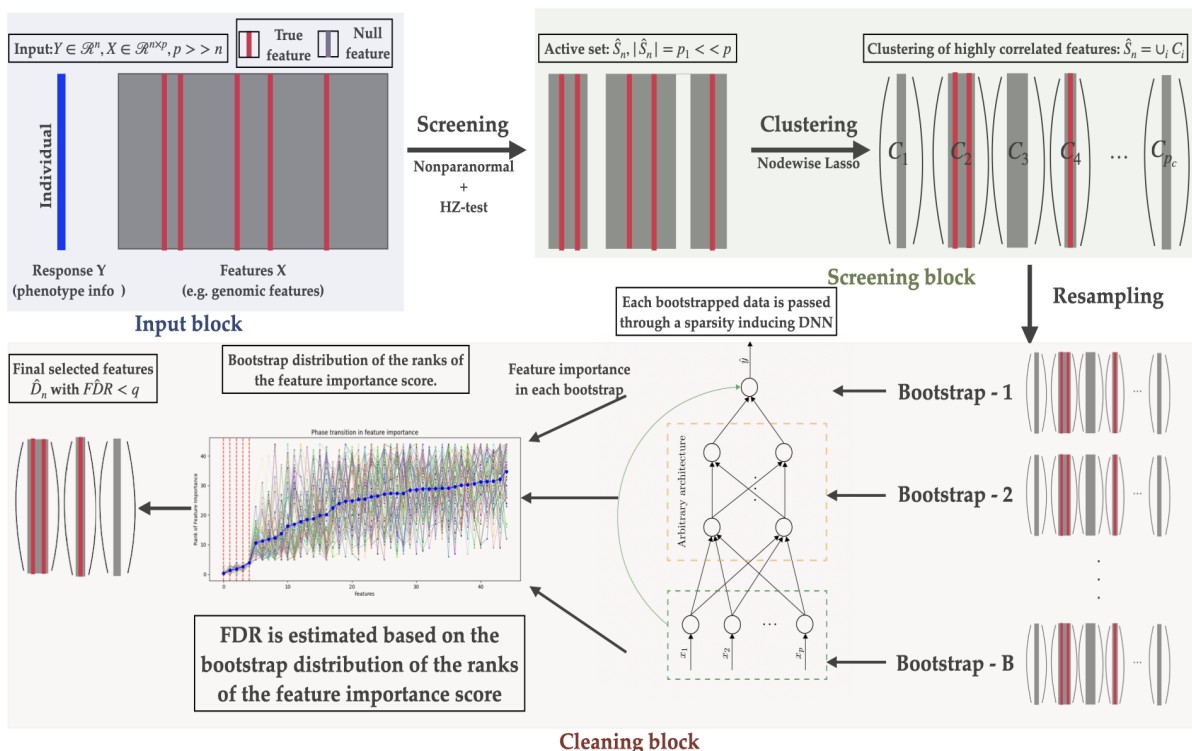

Figure 3: **SciDNet Architecture Overview:** The schematic diagram illustrates the proposed method SciDNet, where the green and red modules represent the screening and cleaning blocks, respectively.

**Our contribution** To address the aforementioned complexities and unexplored gaps in variable selection, particularly when applying deep learning (DL) methods, we introduce the **SciDNet- Screening & Cleaning Incorporated Deep Neural Network**. This novel approach enables reproducible high-dimensional nonlinear-nonparametric feature selection in the presence of highly correlated predictors. The method consists of two steps: screening and cleaning. In the screening step, we perform dimension reduction by eliminating null features and constructing a set of multi-resolution clusters. These clusters collectively encompass all proxy variables necessary to capture the truly significant features with a high probability. In the cleaning step, we employ a carefully tuned DL model with an appropriate resampling scheme to estimate the False Discovery Rate (FDR). This estimation serves as a key component for selecting clusters of highly correlated predictors while controlling the FDR. To this end, 'FDR observed for SciDNet' would implicitly refer the value of the cFDR discussed above. Our major contributions can be summarized below:

- The proposed method, SciDNet, incorporates techniques from statistical machine learning and sparse modeling with the DL framework. It introduces a resampling-based FDR estimation scheme, allowing for the identification of relevant features while discarding irrelevant ones in an FDR-controlled setting. Moreover, SciDNet is specifically designed to handle the challenges posed by highly correlated features, which are known to be problematic for traditional feature selection methods.

- To the best of our knowledge, no other method in the literature addresses the multicollinearity issue in a high-dimensional setting through data adaptive cluster formation, followed by nonlinear-nonparametric error-controlled feature selection integrated with DL. Extensive simulations and real data analyses demonstrate the validity and efficacy of our proposed method. Additionally, an ablation study dissects the contributions of individual steps in the overall performance of the feature selection method.

- Unlike existing state-of-the-art methods, our approach relies on minimal modeling assumptions and does not rely on p-values. As such, it provides a robust and better understanding of the sparse

relationship between the outcome and the high-dimensional predictors. Through theoretical analysis, we establish the provable FDR control guarantee of SciDNet in an asymptotic setting, further consolidating the empirical results.

Overall, our contributions provide a powerful tool for researchers and practitioners grappling with the challenge of selecting relevant variables from highly correlated ultrahigh-dimensional datasets across various fields, ranging from biology to finance to social sciences. The rest of the article includes a description of the proposed screening and cleaning method in Section 2, followed by an extensive simulation study in Section 3, and analyses of two real-world gene expression datasets in Section 4. Section 5 concludes with a summary of findings and future directions. Theoretical study and additional simulation results are relegated to the appendix.

## 2 The Algorithm

### 2.1 Notation and assumptions

Under the supervised learning framework, let $Y$ denote a continuous response variable, and $X = (X_1, \ldots, X_p)$ denote p continuous covariates. Let $F_y(\cdot)$ denote the CDF of the response variable $Y$, and let $F_k(\cdot)$ denote the CDF of the predictor $X_k$. The dataset comprises $n$ independent and identically distributed (iid) samples, each represented as $(Y_i, X_i) = (Y_i, X_{1i}, X_{2i}, \ldots, X_{pi})$, where $i$ refers to the data point index. Notably, we focus on the ultrahigh-dimensional setting where the number of predictors, $p$, grows exponentially with $n$ as $p = O(exp(n^\tau))$, with $\tau > 0$. We assume no specific functional relationship between the outcome $Y$ and the predictors $X$ but we impose a high-level assumption on the distribution of X. In particular, we assume that the predictors follow the nonparanormal distribution; i.e., there exist unknown differentiable functions $g(X) = \{g_j(X_j), j \in \{1, 2, \ldots, p\}\}$, such that $g(X) \sim N(\mu^{p \times 1}, \Sigma^{p \times p})$. This nonparanormal distribution encompasses a wide range of parametric families of distributions and maintains the conditional dependency structure of the original variables $X$. More theoretical details on the nonparanormal family of distributions can be found in Liu et al. (2009). Additionally, to enforce sparsity, we assume the existence of a subset $S_0 \subset 1, 2, \ldots, p$ with $|S_0| = O(1)$, such that, conditional on the features in $S_0$, the response $Y$ is independent of the features in $S_0^c$. In other words, $S_0 = \{k : f(y|X) \text{ depends on } X_k\}$, where $f(y|X)$ is the conditional density of $y$ given $X$. Our objective is to learn the sparsity structure by estimating $S_0$.

### 2.2 Screening Step

Under the assumption that the cardinality of $S_0$ is much smaller than the feature space dimension $p$, most of the features belong to $S_0^c$. Hence in the screening step, we focus primarily on finding an active set $\hat{S}_n$ with $|\hat{S}_n| << p$ such that $P(S_0 \subset \hat{S}_n) \to 1$ as $n \to \infty$. This property is called the sure screening property (Fan & Lv, 2008), which ensures that all the significant predictors are still retained in $\hat{S}_n$ and the other predictors $\{X_j, j \in \hat{S}_n^c\}$ are henceforth eliminated from the remaining analysis. As these active variables are highly correlated among themselves, in the second step we further cluster them by exploiting the conditional dependency structure.

#### 2.2.1 Finding the active set of variables

To find the active set, we first consider the nonparanormal transformation on $(Y, X)$ and then perform the Henze–Zirkler's (HZ) test on the transformed variable. While the first transforms all the variables to a joint Gaussian variable maintaining their conditional covariance structure; the second test confirms, by pairwise testing, if there is significant dependence in the transformed response and predictors. This workflow has been proposed by Liu et al. (2009),Henze & Zirkler (1990), Xue & Liang (2017). The strategy proceeds as follows:

1. **nonparanormal transformation**: We first consider the following transformation: $T_y(Y) = \mathbf{\Phi}^{-1}(F_y(Y)), T_k(X_k) = \mathbf{\Phi}^{-1}(F_k(X_k)), k = 1, 2, \ldots, p$, where $\mathbf{\Phi}(\cdot)$ denotes the CDF of the standard Gaussian distribution. However, in practice, the cdf of $Y$ and $X_k$ are unknown, we can estimate it by the truncated empirical cdf as suggested by Liu et al. (2009). Henceforth, let $(\tilde{T}_y(Y), \tilde{T}_k(X_k))$ denote the corresponding transformations.

2. **HZ test**: By the basic properties of CDF, it is easy to see that $(T_y(Y), T_k(X_k))$ will jointly follow a bivariate Gaussian distribution $N_2(0, I_2)$ if and only if $Y$ is independent of $X_k$. This can be tested using HZ test, Henze & Zirkler (1990), where the test statistic for the predictor $X_k$ can be expressed as $w_k = \int_{\mathcal{R}^2} |\psi_k(t) - exp(-\frac{1}{2}t't)|^2 \phi_\beta(t)dt, k = 1, 2, \ldots, p$; where $\psi_k(t)$ is the characteristic function of $(T_y(Y), T_k(X_k))$ and $exp(-\frac{1}{2}t't)$ represents the characteristic function of $N_2(0, I_2)$. It typically measures the disparity between the joint distribution of $(T_y(Y), T_k(X_k))$ and $N_2(0, I_2)$ and is expected to be typically high for the non-null predictors $X_j, j \in S_0$ indicating significant evidence against the independence of the transformed variable $(T_y(Y), T_k(X_k))$.

Next, as in practice, we proceed with $(\tilde{T}_y(Y), \tilde{T}_k(X_k))$, we calculate the the HZ test statistic as

$$\tilde{w}_k^* = \frac{1}{n^2} \sum_{i=1}^n \sum_{j=1}^n e^{-\frac{\beta^2}{2}d_{ij}} - \frac{2}{n(1+\beta^2)} \sum_{i=1}^n e^{-\frac{\beta^2}{2(1+\beta^2)}d_i} + \frac{1}{1+2\beta^2} \qquad (1)$$

where $d_{ij} = (\tilde{T}_k(x_{ki}) - \tilde{T}_k(x_{kj}))^2 + (\tilde{T}_Y(y_i) - \tilde{T}_Y(y_j))^2$ and $d_i = \tilde{T}_k^2(x_{ki}) + \tilde{T}_Y^2(y_i)$. Consistent with the existing literature, we choose the value of the smoothing parameter $\beta$ as $\frac{(1.25n)^{1/6}}{\sqrt{2}}$, which corresponds to the optimal bandwidth for a nonparametric kernel density estimator with Gaussian kernel (Henze & Zirkler (1990)). The observed test statistics $\tilde{w}_k^*$ converges to $w_k$ as shown in Xue & Liang (2017).

3. Next, we select the active set of predictors $\hat{S}_n$ according to the larger values of $\tilde{w}_k^*$, i.e., $\hat{S}_n = \{1 \leq k \leq p : \tilde{w}_k^* > cn^{-\kappa}\}$ where where $c$ and $\kappa$ are predetermined threshold values.

This active set $\hat{S}_n$ contains all the predictors significantly correlated with the response marginally. Under very mild regularity conditions on the signal strength of the nonnull predictors where $\min_{k \in S_0} w_k \geq 2cn^{-\kappa}$ with c as a constant and $0 \leq \kappa \leq \frac{1}{4}$, the screening process enjoys the advantage of *sure screening property*, i.e., $P(S_0 \subset \hat{S}_n) \to 1$, as $n \to \infty$. More details on the theoretical guarantee can be found in Xue & Liang (2017). A common practice is to set the active set size $|\hat{S}_n|$ at $\nu_n = [n/log(n)]$. However, as we further cluster the active variables in the next step, our proposed method is fairly robust in terms of the $|\hat{S}_n|$ as long as we retain most of the significant variables. We propose to select a bigger active set with size proportional to $\nu_n$.

### 2.2.2 Clustering the active predictors using the precision matrix

---

**Algorithm 1:** Finding clusters and the representatives

**Input** : $(X \in \mathcal{R}^{n \times p}, Y \in \mathcal{R}^n)$, The Active set $\hat{S}_n$ , $|\hat{S}_n| = p_1 < p$

1  Estimate the precision matrix: $\hat{\Sigma}^{-1} = (\hat{\sigma}^{ij})_{i,j \in \{1,2,\ldots,p\}}$ using Nodewise Lasso
2  Define the clusters $C_i = \{j \in \hat{S}_n : \hat{\sigma}^{ij} \neq 0\}, i \in \hat{S}_n$
3  **for** $1 \leq i \leq p_1$ **do**
4      **for** $1 \leq j \leq p_1, j \neq i$ **do**
5          Define $\Omega^{ij} = \{corr(X_{C_i}, X_{C_j})\} = \{\rho(X_l, X_{l'}), l \in C_i, l' \in C_j\} \in \mathcal{R}^{|C_i||C_j|}$
6          **if** $\max\{\Omega^{ij}\} \geq r$ **then**
7              $C_i = C_i \cup C_j$
8              $C_j = \phi$
9          **end if**
10     **end for**
11 **end for**
12 Retain only the non-null clusters: $C = \{C_i : C_i \neq \phi, i \in \hat{S}_n\}, |C| = p_c$
13 Find the cluster representatives $\tilde{S}_n = \{R_j, 1 \leq j \leq p_c : R_j = \underset{l \in C_j}{argmax}\{\tilde{w}_l^*\}\}$

**Output :** Clusters $C_1, C_2, \ldots, C_{|C|}$ and corresponding cluster representatives $\tilde{S}_n = \{R_j, 1 \leq j \leq p_c\}$

---

The Henze-Zirkler test focuses on pairwise marginal correlations between predictors and the response variable, often including null predictors that exhibit strong associations with significant predictors. Consequently, these null predictors tend to be highly correlated among themselves. To address this issue, our strategy aims to

reduce the correlation within the active set by leveraging the conditional dependency structure and dividing the active variables $X_j : j \in \hat{S}_n$ into $p_c (<< p)$ non-overlapping clusters denoted as $C_1, C_2, \ldots, C_{p_c}$. The use of a sparse precision matrix, which captures the dependence structure in high-dimensional feature spaces, has been widely recognized in the statistical literature (e.g., Lauritzen (1996), Shojaie & Michailidis (2010)) due to its scalability. By analyzing the conditional dependence structure, we can gain valuable insights that surpass the analysis of a simple covariance matrix. For instance, in the study of the human brain, two distinct regions may exhibit high correlation without a direct relationship, only due to their strong interaction with a common third region. Hence, without considering the conditional dependencies, simple correlation-based clustering would lead to large cluster sizes and less interpretable groups of brain regions (Das et al., 2017).

To this end, in order to estimate the precision matrix we implement the nodewise Lasso algorithm (Van de Geer et al., 2014) on the transformed variables $(\tilde{T}_y(Y), \tilde{T}_k(X_k)), k \in \hat{S}_n$. Nodewise Lasso regression is generally entertained to estimate a sparse precision matrix in the context of the Gaussian graphical model by performing simultaneous Lasso regression on each predictor. The tuning parameters in each nodewise Lasso are typically selected using cross-validation. More details on this algorithm and its theoretical guarantees can be found in (Meinshausen & Bühlmann, 2006). Let $\hat{\Sigma}^{-1}$ be the estimated precision matrix by the nodewise Lasso algorithm and $\rho(Z, Z')$ denotes any correlation metric for two random variables $Z$ and $Z'$, e.g. Pearson's correlation. Algorithm 1 summarizes the clustering step.

Here, in the clustering step, not only we are clustering the active predictors, but also selecting an appropriate representative from each cluster. First, for each active predictor $X_i \in \hat{S}_n$, we collect all the other active predictors conditionally dependent on $X_i$, and make cluster $C_i$. Although clustering using conditional dependence produces smaller clusters, there might be some overlaps owing to the complex association in the original predictor space. Hence, to reduce the excessive intercluster correlation, we merge all those clusters having a maximum correlation greater than some pre-specified threshold r (we typically set r=0.9). Next, each cluster is updated by adding all the other features conditionally dependent on the existing cluster members. Finally, to find the appropriate cluster representatives, we focus on the HZ-test statistic $\tilde{w}_k^*$ in 1 which measures the extent of resemblance between the distribution of each nonparanormally transformed variable in a cluster and the null distribution $N(0, I_2)$. So, for cluster $C_i$, we select the variable $R_i = argmax_{j \in C_i} \{\tilde{w}_j^*\}$ indicating its strongest association with the response variable compared to the other predictors in the cluster.

## 2.3 Cleaning with Deep Neural Network (DNN)

We start the cleaning step by modeling the response $Y$ and the cluster representatives $X_{\tilde{S}_n}$ obtained through 2.2.2. In order to perform the error-controlled variable selection, each representative will be assigned an importance score followed by a resampling algorithm to finally control the FDR.

While it is possible to adopt any other generic sparsity-inducing DNN procedure, here we focus on the LassoNet algorithm recently proposed by Lemhadri et al. (2021) for its elegant mathematical frameworks which naturally sets the stage for nonlinear feature selection. To approximate the unknown functional connection, it considers the class of all fully connected feed-forward residual neural networks; namely, $\mathcal{F} = \{f \equiv f_{\theta,W} : x \mapsto \theta^T x + h_W(x)\}$. Here, $W$ denotes the network parameters, $K$ denotes the size of the first hidden layer, $W^{(0)} \in \mathcal{R}^{p \times K}$ denotes the first hidden layer parameters, $\theta \in \mathcal{R}^p$ denotes the residual layer's weights. In order to minimize the reconstruction error: the LassoNet objective function can be formulated as:

$$\min_{\theta, W} L(\theta, W) + \lambda ||\theta||_1 \text{ subject to } ||W_j^{(0)}||_\infty \leq M|\theta_j|, j = 1, 2, \ldots, p \tag{2}$$

With $L(\theta, W) = \frac{1}{n} \sum_{i=1}^{n} l(f_{\theta,W}(x_i), y_i)$ as the empirical loss on the training data and $x_i$ as the vector of cluster representatives observed for the $i^{th}$ individual. While the main feature sparsity is induced by the $L_1$ norm on residual layer parameter $\theta$, the second constraint controls the total amount of nonlinearity of the predictors. As mentioned in Lemhadri et al. (2021), LassoNet can be argued as an extension of the celebrated Lasso algorithm to nonlinear variable selection.

In $L_1$ penalization framework, the importance of a specific feature is naturally embedded into the highest penalization level up to which it can survive in the model. So, to measure the importance of each representative, the LassoNet algorithm is executed over a long range of tuning parameter $\lambda_1 \leq \lambda_2 \leq \cdots \leq \lambda_r$ on $(Y, X_{\tilde{S}_n})$.

In practice, a small value is fixed for $\lambda_1$ where all the variables are present in the model. Then, we gradually increase the value of the tuning parameter and stop at $\lambda_r$, where no variables are present in the model. Next, the importance score for the $j$-th cluster is defined as $\hat{\lambda}_j$= maximum value of $\lambda$ up to which the $j$-th representative exists in the model, and then the following rank statistic is computed: $\mathcal{I}_j = \sum_{j' \neq j} \mathbb{1}\left(\hat{\lambda}_j \leq \hat{\lambda}_{j'}\right)$ for $j = 1, 2, \ldots, p_c$. A lower $\mathcal{I}_j$ means that the $j$-th cluster representative stays in the model up to a higher value $\lambda$ implying its high potential as a significant cluster. In contrast, a higher $\mathcal{I}_j$ indicates the corresponding cluster leaves the model even for a smaller value of $\lambda$ as a consequence of being simply a collection of null features. Hence, we should only focus on the clusters with lower ranks. Additionally, in order to control the FDR, understanding the behavior of the predictors under the null distribution is important. In traditional FDR controlling algorithms, this is typically done by generating the p-values. Here, as a p-value-free algorithm, we propose the following resampling-based approach:

1. Generate B bootstrap versions of the data $\left\{Y^b, X^b_{\tilde{S}_n}\right\}_{b=1}^B$ considering only the cluster representatives $\tilde{S}_n$. For each bootstrap version, run the LassoNet algorithm parallelly, and calculate the importance of each representative by measuring $\hat{\lambda}^b_j$= maximum value of $\lambda$ up to which the j-th predictor exists in the model for b-th bootstrap version, and then the ranks $\mathcal{I}^b_j = \sum_{j' \neq j} \mathbb{1}\left(\hat{\lambda}^b_j \leq \hat{\lambda}^b_{j'}\right)$.

   Therefore, the averaged rank is: $\bar{\mathcal{I}}_j = \frac{1}{B} \sum_{b=1}^B \mathcal{I}^b_j$

2. For an arbitrary threshold $\delta$, we would select the cluster representatives with averaged rank $\bar{\mathcal{I}}_j$ lower than $\delta$; so we define, $N_+(\delta) = \sum_{j \in \tilde{S}_n} \mathbb{1}(\bar{\mathcal{I}}_j \leq \delta)$ representing the number of selected clusters with respect to the cutoff $\delta$.

3. Next, to estimate the expected number of falsely discovered clusters, define $\mathcal{R}^b = \{j : \mathcal{I}^b_j \leq \delta\}$, the number of cluster representatives with higher importance score so that the corresponding rank is lower than the cutoff $\delta$ in the b-th bootstrap version. Additionally, define a neighbourhood $\mathcal{N}(\bar{\mathcal{I}}_j, \kappa) = \{l \in \{1, 2, \ldots, p_c\} : |\bar{\mathcal{I}}_j - l| \leq \kappa\}$, for some specific small number $\kappa$.

4. Further, we estimate the number of falsely discovered clusters and hence an estimator of the FDR can be constructed as $F\hat{D}R(\delta) = \frac{\hat{e}_0(\delta)}{N_+(\delta)}$ where,

$$\hat{e}_0(\delta) = \frac{2}{B} \sum_{b=1}^B \left\{\sum_{j \in \mathcal{R}^b} \mathbb{1}(\mathcal{I}^b_j \notin \mathcal{N}(\bar{\mathcal{I}}_j, \kappa))\right\} \tag{3}$$

5. The $F\hat{D}R$ is sequentially estimated with $\delta = \bar{\mathcal{I}}_{(1)}, \bar{\mathcal{I}}_{(2)}, \ldots, \bar{\mathcal{I}}_{(p_c)}$ and the optimum threshold is

$$\Delta^* = \max\{\delta > 0 : F\hat{D}R(\delta) < q\}$$

   for some pre-specific FDR control level $q$. The final selected set of clusters with controlled FDR is given by
$$\hat{D}_n = \{C_j, j = 1, 2, \ldots, p_c : \bar{\mathcal{I}}_j \leq \Delta^*\}$$

The proposed method integrates the concept of false discovery through resampling. If a null predictor obtains a relatively high importance score, it is likely due to a specific bootstrap iteration that generates a spurious relationship, which is not consistent across all other bootstraps. Conversely, significant predictors should consistently yield higher importance scores across all bootstrap iterations. Consequently, the variability in importance score rankings will be substantially greater for null predictors compared to their non-null counterparts. This principle, introduced in the statistical literature as bagging methods (Breiman, 1996; Bühlmann & Yu, 2002), aims to reduce variance in black-box models for prediction/estimation. The proposed method leverages this phase transition phenomenon in the feature selection framework to effectively identify false discoveries. An empirical illustration of this phenomenon is presented in Figure 4. Although we employ non-parametric bootstrap for simplicity in this demonstration, other bootstrap techniques, such as multiplier bootstrap or perturbation bootstrap, can be readily applied. The theoretical investigation is provided in Appendix A, where, following the approach of perturbation bootstrap (Ng & Newton, 2022), we utilize random-weighted Group Lasso penalization to emulate the resampling setup.

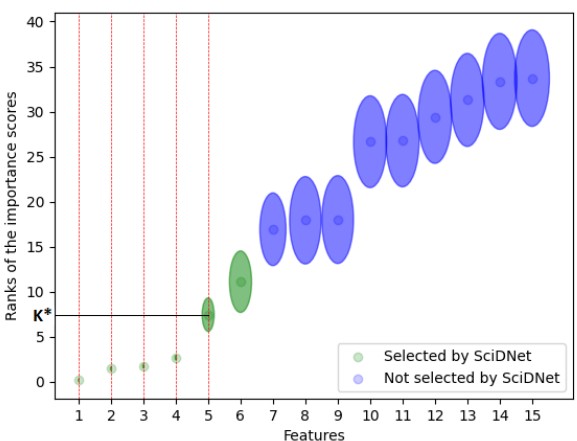

**How to choose the hyperparameter** $\kappa$: Choosing an appropriate value of $\kappa$ has a significant effect on the performance of SciDNet. A higher value of $\kappa$ might lead to weaker control over the inclusion of false discoveries, whereas choosing a small $\kappa$ will create tighter error control resulting in reduced power. However, we propose an effective way to tune the $\kappa$ with the assistance of phase transition in the ranks of the importance score $\bar{\mathcal{I}}_j$ of the cluster representatives. For an illustration, in Figure 4 we consider a single index model (see section 3 for more details of the data-generating mechanism), and the features with top 15 importance scores are shown along the x-axis. The first 5 representative features are the only relevant predictors (indicated by the vertical dotted red line). Along the y-axis, the center of the ellipse for each feature represents the rank of

Figure 4: Illustration of phase transition using synthetic data: Features selected by SciDNet at $q = 0.2$ clearly have lower bootstrap variability compared to the other irrelevant features

the importance scores $\bar{\mathcal{I}}_j$ averaged over 50 bootstrap replications and the area of each ellipse represents the bootstrap variability around the averaged score. One would observe a clear phase transition in the bootstrap distribution of the ranks. For the significant features, the ranks are lower with extremely precise estimates. On the other hand, for the rest of the null features, the averaged ranks possess much higher values coupled with huge variability producing bigger ellipses. Hence, for a compact neighborhood $\mathcal{N}(\bar{\mathcal{I}}_j, \kappa)$ to capture only the small variability in the bootstrap ranks of the significant features, we simply fix $\kappa = \mathbf{K^*}$ (in figure 4), the phase transition point for the averaged rank. This phase transition property is further illustrated on a real data in Appendix B.2.

## 3 Numerical Illustrations

In this section, we investigate the finite-sample performance of SciDNet using a wide spectrum of simulation scenarios. We compare SciDNet to several baseline methods which are widely used in practice. We conduct this simulation study on synthetic data generated from a high-dimensional regression problem.

**Data Generation:** We first consider the single index model for the data-generating mechanism, which is a straightforward yet flexible example of nonlinear models. Here the response is related to a linear combination of the features through an unknown nonlinear, monotonic link function, i.e., $y = g(x'\beta) + \epsilon$. We choose the following link functions: $g(x) = \frac{x^3}{10} + 3\frac{x}{10}$.

We set $n = 400$ and $p = 5000$. The coefficients $\beta \in \mathcal{R}^p$ is sparse with the true nonzero locations $S_0 = \{50, 150, 250, 350, 450\}$, $s = |S_0| = 5$, where $\beta_{S_0^c} = 0, \beta_{S_0} \sim N_5(u\beta_0, 0.1I^{5\times5})$ with $u = \{\pm1\}^5$. The value of $\beta_0$ is set at $\beta_0 = 2, 4$ to incorporate varying signal strength. The random error $\epsilon \sim N(0, \sigma^2)$, with three increasing noise level as $\sigma^2 = 1, 5, 10$. The high dimensional predictors are generated from $X \sim N_p(0, \Sigma)$ where the covariance matrix $\Sigma$ is chosen as a Toeplitz matrix with $\Sigma_{ij} = \rho^{|i-j|}$. To check the effect of multicollinearity, we consider three different settings: $\rho = 0.1, 0.5, 0.95$.

**Evaluation Metrics:** Let $\hat{D}_n$ denote the selected set of features by some algorithm, then we use the following three metrics to evaluate the performance of these feature selection algorithms:

1. Power $= \frac{|\hat{D}_n \cap S_0|}{|S_0|}$, the proportion of relevant features that are correctly identified

2. Empirical FDR $= \frac{|\hat{D}_n \cap S_0^c|}{|\hat{D}_n|}$, the proportion of falsely identified features among all the identified features

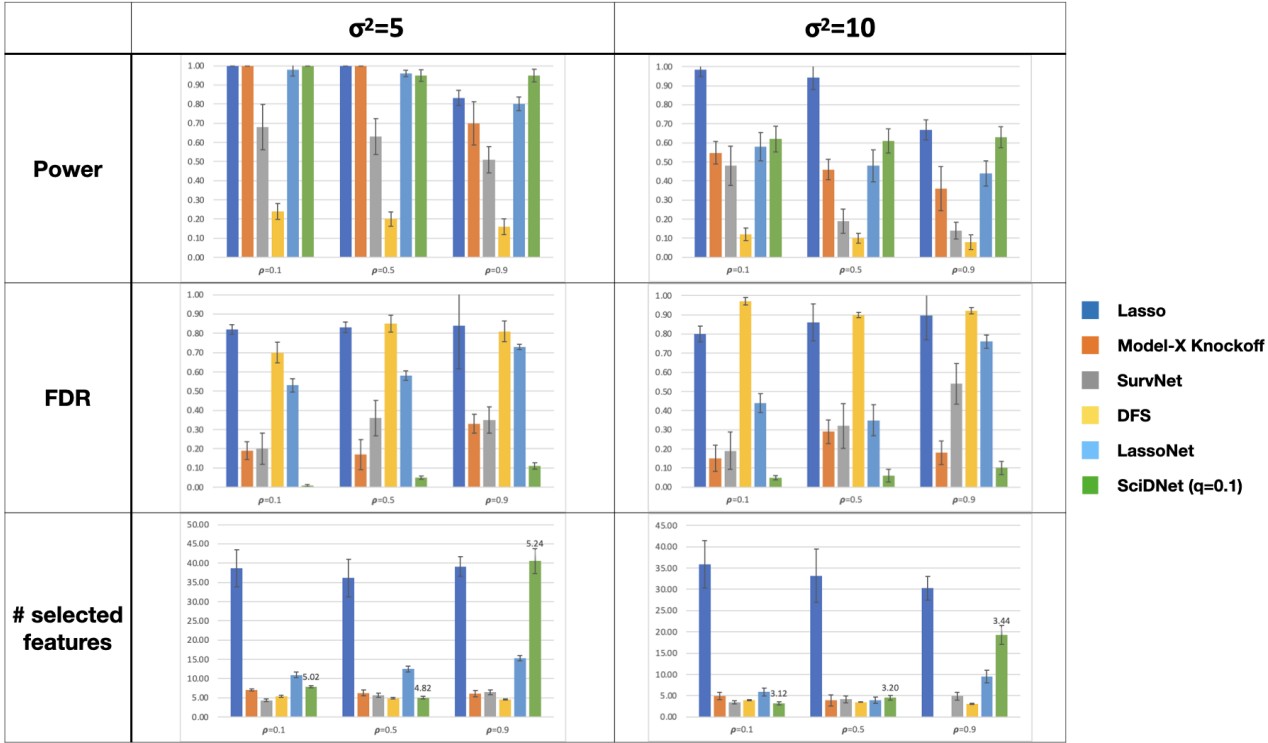

Figure 5: Illustration of the effect of multicollinearity on the feature selection methods

3. $n\_var = |\hat{D}_n|$, the number of total features selected and $n\_clust = |\tilde{S}_n|$, the number of clusters selected by a group feature selection method like SciDnet.

The whole experiment is repeated over 100 Monte Carlo replications and We summarize the results from the empirical evaluations in the following three subsections: (1) The effect of multicollinearity on major existing feature selection methods, (2) the Power vs. FDR balance of SciDNet, and (3) an ablation study showing the effectiveness of all the steps for SciDNet.

### 3.1 The effect of multicollinearity on major existing feature selection methods

We present a simulation study to evaluate the performance of our proposed algorithm in comparison with the following five baseline feature selection methods

1. **Lasso** (Tibshirani, 1996): the $L_1$ penalized linear regression to prevent overfitting and improve model interpretability.

2. **Model-X knockoff** (Candès et al., 2016): A theoretically guaranteed statistical method for FDR control used in high-dimensional variable selection. It constructs "knockoff" variables that mimic the correlations between the original variables and their relationship with the response variable, allowing for control of the false discovery FDR while identifying important variables.

3. **SurvNet** (Song & Li, 2021): A DNN-based FDR control method for feature selection applicable to high-dimensional large datasets.

4. **Deep Feature Selection (DFS)** (Chen et al., 2021): A novel DNN method for feature selection in a high-dimensional setting with complex nonlinear relationships utilizing $L_0$ penalization.

5. **LassoNet** (Lemhadri et al., 2021): The nonlinear extension of Lasso. It combines the advantages of both $L_1$ penalization and neural network structures to identify the important features.

Although several other existing methods exist in the literature for ultra-high dimensional feature selection, we choose these five baseline methods because of their wide applicability and reliable theoretical guarantees. Also, these five methods can be thought of as representative of different classes of algorithms. For example, Model-X knockoff represents the big class of knockoff-based algorithms; whereas SurvNet, DFS, and LassoNet show the effectiveness of the DL-based feature selection methods. Among these five baseline methods, Model-X knockoff and SurvNet are designed to control the FDR and we use $q = 0.2$ as the FDR-control threshold. Also, Lasso, LassoNet, and DFS require proper tuning for their $L_1$ or $L_0$ penalty parameters, which can be done via a grid search. For this purpose, we optimize a BIC-type criterion (Chen & Chen, 2008) to tune these hyperparameters, as suggested by the authors of Chen et al. (2021). For the practical implementation of these baseline methods, we used the code/hyperparameters provided by the authors in the respective papers.

Figure 5 demonstrates the Power, FDR, and the *n_var* of all these methods under different correlation strengths and varying noise levels, fixing $\beta_0 = 2$. Also, as SciDNet selects the features as clusters, we show the *n_clust* in numbers along with *n_var*, in the third row of Figure 5. One would observe that the baseline methods perform poorly as the autocorrelation increases; they result in reduced power and inflated false discoveries. The FDR-controlling methods such as Model-X knockoff and SurvNet fail to control their FDR under the pre-specified threshold $q = 0.2$ for higher autocorrelation. This is somehow expected, as these methods are not tailored for handling such huge multicollinearity. This demonstration empirically motivates as well why we need a feature selection method designed for highly correlated feature space. The DL-based method like LassoNet, DFS, or Survnet additionally suffers from insufficient data under the current big-p-small-n setting. Our additional experiments, presented in Appendix D, demonstrate how these DL-based models gain better power-FDR balance given sufficient training data and moderate correlation among the features. Compared to these baseline methods, SciDNet successfully maintains its power while controlling the FDR below the pre-specified threshold $q = 0.2$ irrespective of the correlation strength. Also, under moderate multicollinearity, when there is no need for clustering, most of the selected clusters by SciDNet are just singleton sets. On the other hand, under the setting of excessive multicollinearity, the individual features become almost indistinguishable. SciDNet addresses this added uncertainty by selecting larger clusters for higher autocorrelation, as demonstrated in the third row of Figure 5.

## 3.2 The Power vs. FDR balance of SciDNet

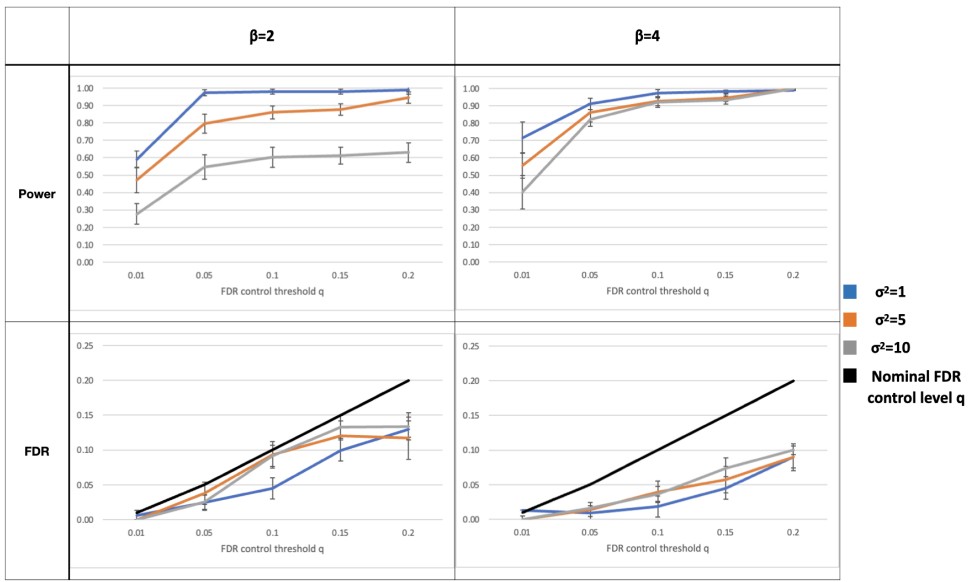

Figure 6: Power vs FDR balance for SciDNet

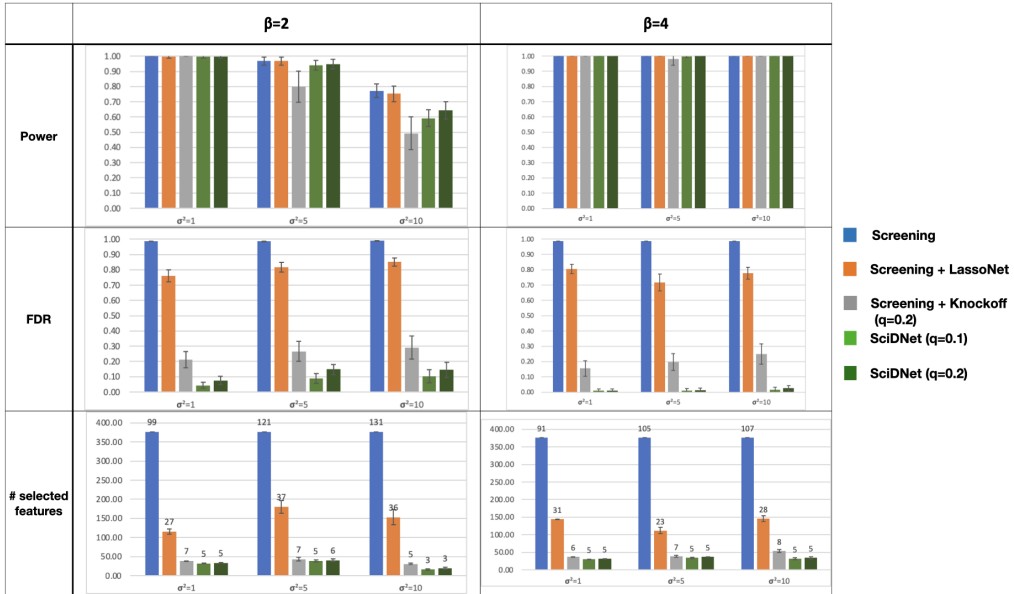

Figure 7: Ablation study for SciDNet

One major validation for an FDR controlling method is to check how it retrieves its power with respect to gradually increasing the FDR-control threshold $q$. Figure 6 shows this power-FDR trade-off for SciDNet. Also, Section 3.1 demonstrates that the baseline methods are not well suited for highly correlated feature space. As SciDNet is specifically designed for this setting, for ease of illustration, we only present the performance of SciDNet setting $\rho = 0.95$. Figure 6 illustrates how our method enjoys quick recovery in power when we gradually increase the FDR-controlling threshold $q$ from 0.01 to 0.20 and also maintain the number of false discoveries below the required level. This illustration empirically validates SciDNet as an FDR-controlled feature selection method.

### 3.3 An ablation study

The proposed method SciDNet is a multi-step process. Its screening step reduces the dimension while retaining the main important features and clustering the highly correlated features. Following this, the cleaning step further uses a sparsity-inducing DL model and finally selects some cluster of features by controlling the FDR via resampling. We aim to analyze the impact of these two steps on the overall performance of the model by adding them one by one and observing the change in the evaluation metrics. We also compare our proposed cleaning step (i.e. resampled LassoNet with FDR estimation) with other possible alternatives like using knockoffs for the cleaning.

Hence, for the ablation study, we compare the following four methods under varying signal and noise strength:

1. **Screening only**,

2. **Screening+LassoNet**: Here we consider LassoNet as the cleaning step

3. **Screenning+Knockoff**: Here we consider Model-X knockoff as the cleaning step, with the FDR-controlling threshold $q = 0.2$

4. **Screening+resampled LassoNet**, i.e. **SciDNet**. We use two FDR-controlling thresholds $q = 0.1$, and 0.2.

The power, empirical FDR, *n_var* are illustrated in Figure 7. In addition, the selected number of clusters, i.e., *n_clust*, are written in numbers in the third row of Figure 7. it empirically consolidates several interesting

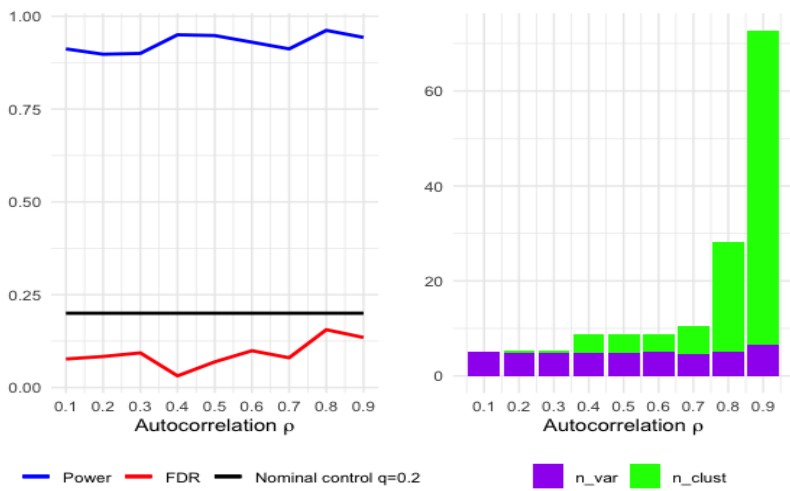

Figure 8: Adaptive cluster formation by SciDNet

characteristics: (a) Due to the sure-screening property 2.2.1, the screening step selects a slightly bigger set of features resulting in high power and high FDR which necessitates further cleaning; (b) All the alternative cleaning steps aim to eliminate the null features further and reduce the FDR while maintaining the power of the screening step; (c) For higher signal and low noise case, SciDNet is comparable to other alternatives. However, for difficult scenarios like low signal and high noise case, which is common in modern genomic and imaging datasets, SciDNet maintains its performance. One would notice that Model-X knockoff loses its FDR control at the nominal level $q = 0.2$ and results in inflated FDR in the presence of high noise. On the other hand, by effectively using the added information from the resampling, SciDNet achieves the best performance (in terms of the power-FDR tradeoff) and continues to preserve its FDR below the nominal level $q$.

Overall, our ablation study showed that the proposed cleaning following the screening step contributes to the overall performance of SciDNet and that removing any of them leads to a significant drop in accuracy. The screening helps in reducing the dimension of the feature space by removing a large chunk of irrelevant features, which further makes the cleaning step computationally very efficient. We also conducted a further simulation study considering several nonlinear models as data-generating processes with Gaussian and non-Gaussian features. The results are presented in Appendix C which further substantiates the results mentioned above that SciDNet maintains a satisfactory power-FDR balance for various complicated nonlinear models with and without interaction terms. The hyperparameter selection and further implementation details of SciDNet are relegated to Appendix B.1 and E, respectively.

### 3.4 Data-adaptive cluster formation with varying multicollinearity

An intriguing characteristic of SciDNet lies in its adaptive estimation of cluster size, enabling it to effectively address the additional uncertainty arising from significant multicollinearity. To demonstrate this capability, we employed SciDNet in a scenario where the data-generating mechanism followed a nonlinear additive model, represented as $y = 2X_{50} + 0.5 \exp(X_{150}) + 2X_{250}^3 + 6X_{350} + 2ReLu(X_{450}) + \epsilon$, with $\epsilon \sim N(0, \sigma^2)$. We set $\sigma^2 = 5$, while keeping the rest of the simulation setting consistent with the description at the beginning of Section 3. The autocorrelation coefficients among the features were varied as $\rho = \{0.1, 0.2, \ldots, 0.9\}$.

Figure 8 demonstrates several key aspects of SciDNet's performance. First, with varying levels of multi-collinearity strength, SciDNet effectively maintains a desirable power-FDR tradeoff, successfully controlling the FDR below the nominal level $q = 20\%$. Second, SciDNet adapts to the added uncertainty caused by high multicollinearity by inflating the cluster size. Notably, in the figure, when there is lower auto-correlation and no need for clustering, the number of selected features ($n\_var$) and the number of selected clusters ($n\_clust$) are almost equal, indicating that the majority of selected clusters are single-tone sets. Conversely,

for higher auto-correlation, SciDNet produces larger clusters, enhancing intra-block correlation and reducing inter-block correlation, while keeping the number of clusters ($n\_clust$) nearly constant, thereby achieving a stable power-FDR balance across the entire range of the autocorrelation coefficient $\rho$. This experiment provides additional evidence of the robustness of our proposed method.

Overall, the simulation study provides strong evidence supporting several key aspects of SciDNet, including its effectiveness, ablation, and robustness. Furthermore, due to the independent nature of most of the intermediate steps of SciDNet, parallel execution is possible, rendering it lightweight and computationally feasible even for ultra-high dimensional datasets. As an illustration, for this simulation setting, the complete execution of SciDNet takes approximately 1.56 minutes (61.31 sec for screening and 32.57 sec for cleaning), in contrast to the Model-X knockoff Candès et al. (2016) which requires 7.4 minutes for generating knockoff features in a 2000-dimensional feature space using "SDP"-style knockoff generation (refer to Patterson & Sesia (2022) for more details). Notably, SciDNet eliminates the need for hyperparameter grid search, which can be challenging in practice for ultra-high dimensional datasets. For instance, DFS (Chen et al. (2021)) requires approximately 1.65 minutes for execution with a fixed hyperparameter value, followed by an extensive grid search that inflates computational complexity. All experiments were performed on a high-performing computing facility equipped with Intel(R) Xeon(R) Platinum 8260 CPU @ 2.40GHz and 4 Tesla V100S. The source code is available at an anonymous repository [link: https://anonymous.4open.science/r/SciDNet-9462].

## 4 Real Data Analysis

In addition to the simulation studies, we implemented the proposed algorithm SciDNet in the following two publicly available gene-expression data sets - the CCLE dataset and the riboflavin dataset. We substantiate the findings in two ways: We first provide supporting evidence from the domain research. Additionally, as a more data-aligned validation, we demonstrate that several generic prediction models significantly gain in test accuracy when applied only on the few features selected by SciDNet compared to the prediction result considering the whole feature space. For this purpose, consistent with the other genomic studies, we use the prediction correlation $Corr(Y_{Pred}, Y_{Test})$ in addition to the test MSE as a metric to measure the test performance. To overcome the extra burden of the low sample size and ultrahigh dimensionality in these data sets, we consider 50 independent replications where the data is divided into training and testing maintaining an $8 : 2$ ratio to get the metrics for the test performance. The final estimate is obtained by averaging all the test MSEs calculated on each of these replications. A similar approach is considered for the correlation metric as well.

### 4.1 Selection of Drug Sensitive Genes using CCLE dataset

A recent large-scale pharmacogenomics study, namely, the cancer cell line encyclopedia (CCLE, link available here), investigated multiple anticancer drugs over hundreds of cell lines. Its main objective is to untangle the response mechanism of anticancer drugs which is critical to precision medicine. The data set consists of dose-response curves for 24 different drugs across over $n = 400$ cell lines. For each cell line, it consists of the expression data of $p = 18,926$ genes, which we consider as features. For the response, we used the area under the dose-response curve, known as activity area in (Barretina et al., 2012), as the response variable to measure drug sensitivity. The activity area represents the extent to which a drug enters a person's bloodstream after administration and is a continuous variable. Here, in this ultra-high dimensional regression setting, we seek to uncover the set of genes associated with the following five specific anticancer drug sensitivity: Topotecan, 17-AAG, Irinotecan, Paclitaxel, and AEW541. These drugs have been used to treat ovarian cancer, lung cancer, and other cancer types. Previous research outputs on these drugs and related gene expression data can be found elsewhere (Barretina et al., 2012).

SciDNet produces multi-resolutional clusters of genes for each of the five drugs considered which are interpretable from the domain science perspective. For example, SciDNet discovers *SLFN11* as the top drug-sensitive gene for the drugs Topotecan and Irinotecan. This is consistent with the previous findings as Barretina et al. (2012); Zoppoli et al. (2012) reported the gene *SLFN11* to be highly predictive for both drugs. For another drug 17-AAG, SciDNet discovers the gene *NQO1* as the topmost important gene which is known to be highly sensitive to 17-AAG (Hadley & Hendricks, 2014). The full table containing all the genes

Table 1: Drug-sensitive genes identified by SciDNet and related prediction performance

| Drug | # genes (clusters) selected | | Test MSE | | | Corr($Y_{Pred}, Y_{Test}$) | | |
|---|---|---|---|---|---|---|---|---|
| | by SciDNet | by LassoNet | LassoNet | SciDNet + MLP | SciDNet + RT | Lassonet | SciDNet + MLP | SciDNet + RT |
| Topotecan | 25 (9) | 18469 | 1.25 (0.21) | 1.23 (0.14) | 0.81 (0.16) | 0.47 (0.11) | 0.58 (0.06) | 0.69 (0.07) |
| 17-AAG | 12 (8) | 7152 | 1.04 (0.16) | 1.05 (0.09) | 0.83 (0.15) | 0.20 (0.16) | 0.33 (0.10) | 0.49 (0.10) |
| Irinotecan | 18 (7) | 17727 | 0.93 (0.20) | 1.09 (0.18) | 0.61 (0.13) | 0.59 (0.10) | 0.63 (0.07) | 0.73 (0.08) |
| Paclitaxel | 18 (8) | 16437 | 1.46 (0.33) | 1.46 (0.23) | 1.11 (0.24) | 0.44 (0.14) | 0.45 (0.11) | 0.59 (0.09) |
| AEW541 | 12 (10) | 15145 | 0.33 (0.06) | 0.39 (0.09) | 0.27 (0.05) | 0.30 (0.14) | 0.49 (0.10) | 0.47 (0.12) |

selected by SciDNet at $q = 15\%$ error-control level and relevant findings from previous genomic studies have been relegated to Appendix F.

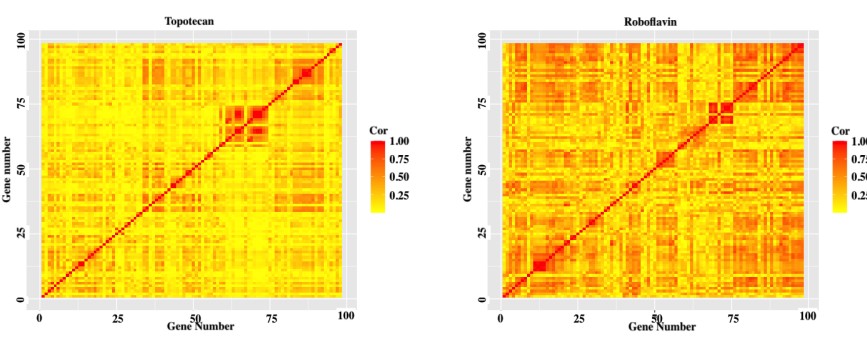

Figure 9: A snapshot of correlation strength for first 100 genes considered for the drug Topotecan in CCLE dataset (left) and riboflavin dataset (right)

Additionally, as in a real setting, it is difficult to check the performance of a feature selection algorithm, here we present a more data-oriented statistical evaluation for further endorsement of SciDNet's discoveries. From the prediction aspect, one would expect that a prediction model implemented only on a handful of features selected by a successful feature selection algorithm would maintain the similar performance of a model implemented on the whole feature space; in some cases, it might enhance the accuracy. To validate this, we randomly split the whole data into 8:2 for training and testing. First, the SciDNet is implemented in the training part, and then two separate prediction models are used only focusing on the selected features : (1) an MLP - a feed-forward multi-layer perceptron with two hidden layers and (2) bagged regression tree (Breiman, 1996). These two experiments are henceforth called: **"SciDNet+MLP"** and **"SciDNet+RT"**. Next, the test data is used to check the out-of-sample prediction accuracy. Furthermore, similar to the simulation study in section 3, we separately implement the LassoNet on the training data for its simultaneous sparsity-induced prediction-optimal characteristics. The summary of the results is presented in table S5 which indicates several interesting points. First, to get the prediction optimal result, LassoNet fails to capture the sparsity and discovers a huge number of genes. This is somehow expected as most of the prediction-optimal sparse methods tend to select a larger set of features to maintain the prediction quality (Wasserman & Roeder, 2009). On the other hand, SciDNet produces only $\sim 10$ clusters with an average cluster size $\sim 2.5$. Even with this huge dimension reduction, the added gain in test MSE and *Corr($Y_{Pred}, Y_{Test}$)* further proves that ScidNet successfully retains all the significant predictors. One would further notice, SciDNet+RT achieves the best stable performance which is consistent for all the drugs and our simulation study as well. This experiment demonstrates that a black-box predictive model produces more accurate results when applied on the features selected by SciDNet rather than its implementation on the whole feature space, indicating SciDNet's potential use in both feature selection and prediction.

## 4.2 Selection of associated genes in Riboflavin production data set

We further implement the SciDNet in the context of riboflavin (vitamin B2) production with bacillus subtilis data, a publicly accessible dataset available in the 'hdi' package in R. Here the continuous response is the logarithm of the riboflavin production rate, observed for $n = 71$ samples along with the logarithm of the expression level of $p = 4088$ genes which are treated here as the predictors. Unlike the previous CCLE data, significant multicollinearity is present in most of the Riboflavin data set, as demonstrated in Figure 9. Hence,

to determine which genes are important for riboflavin production, SciDNet resulted in finding 9 clusters of a total of 160 correlated genes at the $q = 15\%$ FDR control level, making the average cluster size of $\sim 17.78$, which is much bigger compared to the previous analysis. SciDNet discovered the gene *YCIC_at* as one of the expressive genes related to riboflavin production which was identified by Bühlmann et al. (2014) as a causal gene in this context. The full list of the selected cluster of genes by SciDNet is relegated to Appendix F.2.

The results from the empirical validation of SciDNet's feature selection on the riboflavin dataset are presented in table 2. However, for the empirical evaluation, SciDNet+MLP is performing poorly as the inflated cluster dimensions make the input layer of the MLP comparatively large where the number of the training data point is $\approx 57$. This necessitates the need for a sparse model here, and we adopt the idea of *Relaxed Lasso*, first proposed by Meinshausen

Table 2: Prediction performance of SciDNet for Riboflavin production data set

| Algorithm | Test MSE | Corr($Y_{Pred}, Y_{Test}$) |
|---|---|---|
| LassoNet | 0.83 (0.18) | 0.36 (0.30) |
| SciDNet + LassoNet | 0.19 (0.15) | 0.89 (0.12) |
| SciDNet + RT | 0.42 (0.16) | 0.74 (0.15) |
| SciDNet + MLP | 2.64 (0.79) | 0.28 (0.37) |

(2007). Here we implement the LassoNet again on the selected features by SciDNet, which certainly improves the prediction accuracy. Consistent with the previous experiments, SciDNet+RT effectively maintains its prediction performance. This further consolidates the need for applying an apt feature selection method before fitting a predictive model for an explainable research outcome.

## 5 Discussion

While the explainable AI is the need of the hour, statistical models coupled with cutting-edge ML techniques have to push forward because of their solid theoretical foundation clipped with principled algorithmic advancement. The proposed method SciDNet efficiently exploits several existing statistics and ML literature tools to circumvent some of the complexities that simple adaptation of current DL-based models fail to address appropriately. The basic intuition and exciting empirical results of SciDNet on simulated and real datasets open avenues for further research. For example, one may be interested in developing a theoretical foundation for this 'screening' and 'cleaning' strategy for provable FDR control. It would be worth mentioning that although we used the sure independence screening with HZ-test and LassoNet as the main tools, SciDNet puts forward a more generic framework and can be implemented with any other model-free feature screening method and sparsity-inducing DL algorithms like Feng & Simon (2017). In the screening part, a further methodological extension would consider relaxing the assumption of nonparanormally distributed features for a more flexible approach. Additionally, as the dimensionality is reduced after the screening step, it would be interesting to implement model-free knockoff generating algorithms like Romano et al. (2020) in the cleaning step as further algorithmic development. One limitation is that we mainly focus on the regression setup with the continuous outcome because of the requirements of the HZ sure Independence test used in the screening step. For a classification task, any model-free feature screening method like Zhou & Zhu (2018) can be applied in a more general framework.

**Contents of the appendix:** In the appendix, we provide a theoretical investigation showing the conditional convergence of the estimated FDR to an upper bound of the actual FDR. Results from an additional simulation study showing the finite sample performance of the proposed method SciDNet and its comparison with other existing methods and an ablation study showing the effectiveness of different screening and cleaning strategies are presented. The phase transition property of the feature importance is demonstrated by synthetic and real data analysis. Lastly, the model implementation details, link to the code and a discussion on hyperparameter setting are added in the appendix.

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

# A   Appendix 1. Theoretical study

Although the proposed method SciDNet is demonstrated based on the LassoNet algorithm (Lemhadri et al., 2021), any other sparsity-inducing Deep learning framework can be adopted at the cleaning step of SciDNet. Hence, for the theoretical study, we consider a broader framework with a general analytic DNN. Also, as the screening step is theoretically guaranteed by Xue & Liang (2017), we start the theoretical study directly from the cleaning step, assuming the data is $\{Y^i \in \mathbb{R}, X^i \in \mathcal{X}\}_{i=1}^n \overset{iid}{\sim} P_D$, where $\mathcal{X}$ is a bounded open set in $\mathbb{R}^p$, $p < n$ and the input density $p_x$ is positive and continuous on its domain $\mathcal{X}$. In the spirit of Dinh & Ho (2020), we consider the following general analytic neural network model $f_\alpha(x)$, an L-layer neural network with parameters $\alpha = (t_{in}, T_{in}, t_{out}, T_{out}, S)$ and defined by

- **Input layer:** $h_1(x) = t_{in} + T_{in}x$
- **Hidden layers:** $h_j(x) = \phi_{j-1}(S, h_{j-1}(x), h_{j-2}(x), \ldots, h_1(x)), j = 2, 3, \ldots, L-1$
- **Output layers:** $f_\alpha(x) = h_L(x) = t_{out} + T_{out}h_{L-1}(x)$

with $d_i$ = size of the $i$-th layer, $d_1 = p, d_L = 1$, $T_{in} \in \mathbb{R}^{d_2 \times p}, t_{in} \in \mathbb{R}^{d_2}, T_{out} \in \mathbb{R}^{1 \times D_{L-1}}, t_{out} \in \mathbb{R}$ and $\phi_1, \phi_2, \ldots, \phi_{L-2}$ are analytic functions parameterized by the hidden layers' parameter $S$. This general framework covers a wide range of models, including feed-forward networks, convolutional networks, and a major subclass of residual networks. For the sake of theoretical study, we further assume that (1) the set of all feasible vectors $\alpha$ of the model is a hypercube $\mathcal{W} = [-A, A]^{n_\alpha}$, (2) both $\mathcal{W}$ and $\mathcal{X}$ are bounded, (3) $f_\alpha$ is analytic in the sense that there exist $C_1, C_2 > 0$ such that $|f_\alpha(x)| \leq C_1$, and $||\bigtriangledown_\alpha f_\alpha(x)||_\infty \leq C_2, \forall \alpha \in \mathcal{W}, x \in \mathcal{X}$ and these functions are Lipschitz continuous, and (4) $Y = f_{\alpha^*}(X) + \epsilon$ with $\epsilon \sim N(0, \sigma_\epsilon^2), \alpha^* \in \mathcal{W} = [-A, A]^{n_\alpha}$ and we assume that the "true" model $f_{\alpha^*}(\cdot)$ only depends on $x$ through a subset of significant features $S \in 1, 2, \ldots, p$ while being independent of features in $S^c$, $|S| = s = O(1)$. To this end, a general group Lasso estimator (Feng & Simon, 2017; Dinh & Ho, 2020) has been defined as

$$\hat{\alpha}_n = \arg\min_\alpha \{\frac{1}{n}\sum_{i=1}^n l(\alpha, X^i, Y^i) + \lambda_n \sum_{j=1}^p ||\alpha^{[:,j]}||\} \tag{4}$$

where $l(\alpha, x, y) = (y - f_\alpha(x))^2$ is the square-error loss, $\lambda_n > 0$ is the associated penalty parameter, $||\cdot||$ is the standard Euclidean norm and $\alpha^{[:,j]}$ is the vector of parameters associated with j-th input feature.

Now, we incorporate the notion of resampling/bootstrapping by introducing the random-weighted group lasso defined as follows

$$\hat{\alpha}_n^w = \arg\min_\alpha \{\frac{1}{n}\sum_{i=1}^n W_i l(\alpha, X^i, Y^i) + \lambda_n \sum_{j=1}^p ||\alpha^{[:,j]}||\} \tag{5}$$

where $W_i \overset{iid}{\sim} F_W$, independent of the data distribution $P_D$. The random weighting scheme effectively maintains the flavour of perturbation bootstrap (Das & Lahiri, 2019), thus can be used for uncertainty quantification in the context of sparse models. Ng & Newton (2022) studied this in the context of sparse linear model. Here, we first show that under appropriate random weights, $\hat{\alpha}_n^w$ converges to a set of well-behaved optimal hypotheses in the sense that $\mathcal{K} = \{\alpha \in \mathcal{W} : f_\alpha = f_{\alpha^*} \text{ and } \alpha^{[:,j]} = 0, \text{ for } j \in S^c\}$.

We note that, with the addition of random weights the overall probability measure changed to $P = P_D \times P_W$ where $P_W$ is the probability measure of the triangular array of random weights. We define the following three sigma fields: $\mathcal{F}_n^w = \sigma\{W_1, W_2, \ldots, W_n\}$, $\mathcal{F}_n^x = \sigma\{X_1, X_2, \ldots, X_n\}$, and $\mathcal{F}_n^y = \sigma\{Y_1, Y_2, \ldots, Y_n\}$.

We further define, the risk function $R(\alpha) = E_{P_D}(Y - f_\alpha(X))^2$, the empirical risk function $R_n(\alpha) = \frac{1}{n}\sum_{i=1}^n (Y^i - f_\alpha(X^i))^2$ and the weighted empirical risk function $R_n^w(\alpha) = \frac{1}{n}\sum_{i=1}^n W_i (Y^i - f_\alpha(X^i))^2$. The equivalence class can be expressed as $\mathcal{H}^* = \{\alpha \in \mathcal{W} : R(\alpha) = R(\alpha^*)\}$.

For simplicity, in order to obtain a bounded weight, we assume $F_W$ is such that $P_W(C_2 < W < C_3) = 1$, for some $C_3 > 0$.

**Lemma A.1** (Probabilistic Lipschitzness of the random-weighted empirical risk). *For any $\delta > 0$, there exists $M_\delta > c_0$ such that $R_n^w(\alpha)$ is an $M_\delta$-Lipschitz function with probability at least $1 - \delta$.*

*Proof.* We note that,

$$
\begin{aligned}
|R(\alpha) - R(\beta)| &= |E\left((Y - f_\alpha(X))^2 - (Y - f_\beta(X))^2\right)| \\
&\leq E|\left(f_\alpha(X) - f_\beta(X)\right)\left(2Y - f_\alpha(X) - f_\beta(X)\right)| \\
&\leq C_2||\alpha - \beta||E|\left(2Y - f_\alpha(X) - f_\beta(X)\right)| \\
&\leq C_2||\alpha - \beta||\left(2E|Y - f_{\alpha^*}(X)| + E|\left(f_\alpha(X) + f_\beta(X) - 2f_{\alpha^*}(X)\right)|\right) \\
&\leq C_2||\alpha - \beta||(2\sigma + 4C_1)
\end{aligned}
$$

Similarly,

$$
\begin{aligned}
|R_n^w(\alpha) - R_n^w(\beta)| &= |\frac{1}{n}\sum_{i=1}^n W_i\left((Y^i - f_\alpha(X^i))^2 - (Y^i - f_\beta(X^i))^2\right)| \\
&\leq \frac{1}{n}\sum_{i=1}^n W_i|\left(f_\alpha(X^i) - f_\beta(X^i)\right)\left(2Y^i - f_\alpha(X^i) - f_\beta(X^i)\right)| \\
&\leq C_2||\alpha - \beta||\frac{1}{n}\sum_{i=1}^n W_i\left(2|Y^i - f_{\alpha^*}(X^i)| + |\left(f_\alpha(X^i) + f_\beta(X^i) - 2f_{\alpha^*}(X^i)\right)|\right) \\
&\leq C_2||\alpha - \beta||\left(4c_1 + \frac{2}{n}\sum_{i=1}^n W_i|\epsilon^i|\right)
\end{aligned}
$$

Thus for all $M_\delta > 4C_1C_2$, the proof is completed by noting the following

$$
\begin{aligned}
P\left(|R_n^w(\alpha) - R_n^w(\beta)| \leq M_\delta||\alpha - \beta||, \forall \alpha, \beta \in \mathcal{W}\right) &\geq P\left(\frac{1}{n}\sum_{i=1}^n W_i|\epsilon^i| \leq \frac{M_\delta}{2C_2} - 2C_1\right) \\
&= E_{P_W}\left[P\left(\frac{1}{n}\sum_{i=1}^n W_i|\epsilon^i| \leq \frac{M_\delta}{2C_2} - 2C_1|W_1, W_2, \ldots, W_n\right)\right] \\
&= E_{P_W}\left[1 - P\left(\frac{1}{n}\sum_{i=1}^n W_i|\epsilon^i| \geq \frac{M_\delta}{2C_2} - 2C_1|W_1, W_2, \ldots, W_n\right)\right] \\
&\geq E_{P_W}\left[1 - \frac{\frac{1}{n}\sum_{i=1}^n W_i E|\epsilon^1|}{\frac{M_\delta}{2C_2} - 2C_1}\right] \geq 1 - \frac{C_3 E|\epsilon^1|}{\frac{M_\delta}{2C_2} - 2C_1}
\end{aligned}
$$

$\square$

**Lemma A.2** (Generalization bound for the random-weighted empirical risk)**.** *For any $\delta > 0$, $\exists\ C_4(\delta) > 0$ such that* $P\left(|R_n^w(\alpha) - R(\alpha)| \leq C_4\frac{log(n)}{\sqrt{n}}\right) \geq 1 - \delta, \forall \alpha \in \mathcal{W}$

*Proof.* Note that $nR_n^w(\alpha) = \sum_{i=1}^n W_i\left(Y^i - f_\alpha(X^i)\right)^2 = \sum_{i=1}^n W_i Z_i^2$, where $Z_i = Y^i - f_\alpha(X^i) \sim N\left(f_{\alpha^*}(X^i) - f_\alpha(X^i), \sigma_\epsilon^2\right)$, conditional on $\mathcal{F}_X$.

Hence, conditional on $\mathcal{F}_X, \mathcal{F}_W$, $\frac{nR_n^w(\alpha)}{\sigma_\epsilon^2} \sim$ a weighted non-central $\chi^2$ distribution. We will use the following Lemma 3 to get a sharp tail bound for weighted non-central $\chi^2$ distribution. Hence,

$$
\begin{aligned}
P\left(|R_n^w(\alpha) - R(\alpha)| > \frac{t}{2}\right) &\leq E_{\mathcal{F}_X, \mathcal{F}_W}\left[2exp\left(-\frac{Cn^2t^2}{2n + 2\sum_{i=1}^n W_i\left(f_{\alpha^*}(X^i) - f_\alpha(X^i)\right)^2 - \sum_{i=1}^n W_i}\right)\right] \\
&\leq E_{\mathcal{F}_X, \mathcal{F}_W}\left[2exp\left(-\frac{Cn^2t^2}{2n + 2\sum_{i=1}^n W_i\left(f_{\alpha^*}(X^i) - f_\alpha(X^i)\right)^2}\right)\right] \\
&\leq E_{\mathcal{F}_X, \mathcal{F}_W}\left[2exp\left(-\frac{Cn^2t^2}{2n + 2nC_3 4C_1^2}\right)\right] \leq 2exp(-\tilde{c}nt^2)
\end{aligned}
$$

The rest of the proof follows the direct proof of the Lemma 3.3 Dinh & Ho (2020). ∎

**Theorem A.3** (Convergence of random-weighted Group Lasso)**.** *For any $\delta > 0$, there exist $C_\delta, C' > 0$ and $N_\delta > 0$ such that for all $n \geq N_\delta$,*

$$d(\hat{\alpha}_n^w, \mathcal{H}^*) \leq C_\delta \left( \lambda_n^{\frac{\nu}{\nu-1}} + \frac{logn}{\sqrt{n}} \right)^{\frac{1}{\nu}} \text{ and } ||\hat{\alpha}_n^{w[:,S^c]}|| \leq 4C_4 \frac{logn}{\lambda_n \sqrt{(n)}} + C'd(\hat{\alpha}_n, \mathcal{H}^*)$$

*where $d(x, Z) = \inf_{z \in Z} ||x - z||$.*

The proof directly follows from the above mentioned Lemma 1,2 and the theorem 3.3 from Dinh & Ho (2020).

Theorem A.3 demonstrates the convergence of the random-weighted group-Lasso estimates to a set of "well-behaved" optimal hypotheses $\mathcal{K} = \{\alpha \in \mathcal{W} : f_\alpha = f_{\alpha^*} \text{ and } ||\alpha^{[:,S^c]}|| = 0\} \subset \mathcal{H}^*$. However, this depends on the regularization parameter $\lambda_n$; finding its optimum value is generally a daunting task in practice. As a solution, the importance score considering the whole regularization path would provide a more robust way to identify false discoveries. Following our proposed method in Section 2.3, here we consider the cleaning step in view of the random-weighted group-Lasso penalized DNN. We repeat this step over B independently generated random-weighting scheme and recall the importance scores $\hat{\lambda}_j$ and the corresponding ranks of the importance score $\mathcal{I}_j$ for the b-th scheme as follows:
$\hat{\lambda}_j^b = \max\{\lambda_n \ni ||\alpha_n^{w,[:,j]}|| \neq 0\} = $ maximum value of $\lambda$ up to which the j-th feature exists in the model for the b-th random scheme, and
$\mathcal{I}_j^b = $ rank of the importance scores $= \sum_{j' \neq j} \mathbb{1}\left( \hat{\lambda}_j \leq \hat{\lambda}_{j'} \right)$ and the averaged rank $\bar{\mathcal{I}}_j = \sum_{b=1}^B \mathcal{I}_j^b$

Our basic strategy is to select the features with high-importance scores consistent in all the bootstrap replication. Hence, for a cutoff $\delta$, define the number of selected features as $N_+(\delta) = \sum_{j=1}^p \mathbb{1}\left( \bar{\mathcal{I}}_j \leq \delta \right)$. We further propose our estimate of the FDR as

$$F\hat{D}R(\delta) = \frac{\hat{e}_0(\delta)}{N_+(\delta)},$$

$$\text{where } \hat{e}_0(\delta) = \text{estimated number of false discoveries} = \frac{2}{B} \sum_{b=1}^B \left\{ \sum_{j \in 1}^p \mathbb{1}(\mathcal{I}_j^b \leq \delta, \mathcal{I}_j^b \notin \mathcal{N}(\bar{\mathcal{I}}_j, \kappa)) \right\}$$

We further calculate the data-dependent optimum cutoff as $\Delta^* = \max\{\delta \in \{\bar{\mathcal{I}}_1, \bar{\mathcal{I}}_2, \ldots, \bar{\mathcal{I}}_p\} : F\hat{D}R(\delta) < q\}$ for some pre-specific FDR control level $q$.

**Theorem A.4** (Convergence of the estimated FDR)**.** *Using the data-dependent cutoff $\Delta^*$, the actual FDR is bounded by the user-specified level $q$; that is $E\left( \frac{\hat{e}_0(\Delta^*)}{N_+(\Delta^*)} \right) < q$ as $n \to \infty$.*

*Proof.* The feature importance scores $\bar{\mathcal{I}}_j$ evidently provide the information of the survival of the feature $X_j$ over the whole regularization path. Now, under the setup of the random-weighted group-lasso framework, this is uniquely monitored by the KKT condition: $||\alpha^{[:,j]}|| = 0$ if $||\left( \frac{\partial f_\alpha(X)}{\partial \alpha^{[:,j]}} \right)^T_{p_j \times n} diag(W_1, W_2, \ldots, W_n)(Y - f_\alpha(X))_{n \times 1}|| < \lambda \sqrt{p_j}$, where $p_j$ is the number of total parameters associated with the j-th feature. Hence, this $L_2$ norm in the KKT condition can be treated as the importance score $\hat{\lambda}_j$, mentioned in Section 2.3. Now, for the null features $j \in S_0^c$, the derivative term is $o\left( n^{-1/4}log(n) \right)$ by the continuity of $\bigtriangledown_\alpha f_\alpha(x)$ and the convergence of $\hat{\alpha}_n^w$ to $\mathcal{K}$. Also, by Lemma A.2, the residual term is $o\left( n^{-1/2}log(n) \right)$. Also, the random weights $W$'s are bounded in $[C_2, C_3]$. Hence, this new $\hat{\lambda}_j, j = 1, 2, \ldots, p$ become exchangeable asymptotically.

As a consequence of the exchangeable variables, the ranks of the importance scores $\hat{\lambda}_j$ follows uniform distribution asymptotically. Hence, for large $n$, $\mathcal{I}_j^w = \sum_{j' \neq j} \mathbb{1}\left( \hat{\lambda}_j \leq \hat{\lambda}_{j'} \right) \sim U(p - s + 1, p)$. Now, we consider

the following random variable names *False Discovery Proportion (FDP)* whose expectation is the FDR.

$$FDP = \frac{\sum_{j=1}^{p} \mathbb{1}\left(\bar{\mathcal{I}}_j \leq \Delta^*, j \in S_0^c\right)}{\sum_{j=1}^{p} \mathbb{1}\left(\bar{\mathcal{I}}_j \leq \Delta^*\right)}$$

$$= \underbrace{\frac{\frac{2}{B}\sum_{b=1}^{B}\left\{\sum_{j\in 1}^{p}\mathbb{1}(\mathcal{I}_j^b \leq \Delta^*, \mathcal{I}_j^b \notin \mathcal{N}(\bar{\mathcal{I}}_j, \kappa))\right\}}{\sum_{j=1}^{p}\mathbb{1}\left(\bar{\mathcal{I}}_j \leq \Delta^*\right)}}_{\leq q} \cdot \underbrace{\frac{\sum_{j=1}^{p}\mathbb{1}\left(\bar{\mathcal{I}}_j \leq \Delta^*, j \in S_0^c\right)}{\frac{2}{B}\sum_{b=1}^{B}\left\{\sum_{j\in 1}^{p}\mathbb{1}(\mathcal{I}_j^b \leq \Delta^*, \mathcal{I}_j^b \notin \mathcal{N}(\bar{\mathcal{I}}_j, \kappa))\right\}}}_{R(\Delta^*)}$$

The first part of FPR is $\leq q$ by the typical choice of $\Delta^*$ and hence, in order to show the asymptotic FDR control, all we need to show is $\lim_{n\to\infty} E(R(\Delta^*)) \leq 1$.

$$E(R(\Delta^*)) = E\left(\frac{\sum_{j=1}^{p}\mathbb{1}\left(\bar{\mathcal{I}}_j \leq \Delta^*, j \in S_0^c\right)}{\frac{2}{B}\sum_{b=1}^{B}\left\{\sum_{j\in 1}^{p}\mathbb{1}(\mathcal{I}_j^b \leq \Delta^*, \mathcal{I}_j^b \notin \mathcal{N}(\bar{\mathcal{I}}_j, \kappa))\right\}}\right)$$

$$= E\left(\frac{\sum_{j=1}^{p}\mathbb{1}\left(\bar{\mathcal{I}}_j \leq \Delta^*, \{\mathcal{I}_j^b\}_{b=1}^{B} \sim Uniform\right)}{\frac{2}{B}\sum_{b=1}^{B}\left\{\sum_{j\in 1}^{p}\mathbb{1}(\mathcal{I}_j^b \leq \Delta^*, \mathcal{I}_j^b \notin \mathcal{N}(\bar{\mathcal{I}}_j, \kappa))\right\}}\right)$$

$$= E\left(\frac{\sum_{j=1}^{p}\mathbb{1}\left(\bar{\mathcal{I}}_j \leq \Delta^*, \{\mathcal{I}_j^b\}_{b=1}^{B} \sim Uniform, \mathcal{I}_j^b \in \mathcal{N}(\bar{\mathcal{I}}_j, \kappa), \forall b\right)}{\frac{2}{B}\sum_{b=1}^{B}\left\{\sum_{j\in 1}^{p}\mathbb{1}(\mathcal{I}_j^b \leq \Delta^*, \mathcal{I}_j^b \notin \mathcal{N}(\bar{\mathcal{I}}_j, \kappa))\right\}}\right) +$$

$$E\left(\frac{\sum_{j=1}^{p}\mathbb{1}\left(\bar{\mathcal{I}}_j \leq \Delta^*, \{\mathcal{I}_j^b\}_{b=1}^{B} \sim Uniform, \mathcal{I}_j^b \notin \mathcal{N}(\bar{\mathcal{I}}_j, \kappa), \forall b\right)}{\frac{2}{B}\sum_{b=1}^{B}\left\{\sum_{j\in 1}^{p}\mathbb{1}(\mathcal{I}_j^b \leq \Delta^*, \mathcal{I}_j^b \notin \mathcal{N}(\bar{\mathcal{I}}_j, \kappa))\right\}}\right)$$

$$\leq E\left(\frac{2\sum_{j=1}^{p}\mathbb{1}\left(\bar{\mathcal{I}}_j \leq \Delta^*, \{\mathcal{I}_j^b\}_{b=1}^{B} \sim Uniform, \mathcal{I}_j^b \notin \mathcal{N}(\bar{\mathcal{I}}_j, \kappa), \forall b\right)}{\frac{2}{B}\sum_{b=1}^{B}\left\{\sum_{j\in 1}^{p}\mathbb{1}(\mathcal{I}_j^b \leq \Delta^*, \mathcal{I}_j^b \notin \mathcal{N}(\bar{\mathcal{I}}_j, \kappa))\right\}}\right)$$

$$\leq E\left(\frac{2\sum_{j=1}^{p}\mathbb{1}\left(\bar{\mathcal{I}}_j \leq \Delta^*, \mathcal{I}_j^b \notin \mathcal{N}(\bar{\mathcal{I}}_j, \kappa), \forall b\right)}{\frac{2}{B}\sum_{b=1}^{B}\left\{\sum_{j\in 1}^{p}\mathbb{1}(\mathcal{I}_j^b \leq \Delta^*, \mathcal{I}_j^b \notin \mathcal{N}(\bar{\mathcal{I}}_j, \kappa))\right\}}\right)$$

$$\leq E\left(\frac{\frac{2}{B}\sum_{b=1}^{B}\left\{\sum_{j\in 1}^{p}\mathbb{1}(\mathcal{I}_j^b \leq \Delta^*, \mathcal{I}_j^b \notin \mathcal{N}(\bar{\mathcal{I}}_j, \kappa))\right\}}{\frac{2}{B}\sum_{b=1}^{B}\left\{\sum_{j\in 1}^{p}\mathbb{1}(\mathcal{I}_j^b \leq \Delta^*, \mathcal{I}_j^b \notin \mathcal{N}(\bar{\mathcal{I}}_j, \kappa))\right\}}\right) = 1$$

The third inequality is true by the fact that the neighborhood $\mathcal{N}(\bar{\mathcal{I}}_j, \kappa)$ is much smaller than its complement region. The last inequality is true because of the fact that $P(Z_1 \leq z, Z_2 \leq z, \ldots, Z_p \leq z) \leq \frac{1}{p}\sum_{i=1}^{p} P(Z_i \leq z)$, for any set of random variables $Z_1, Z_2, \ldots, Z_p$. Hence, the actual FDR is controlled at the user-specific bound $q$ by selecting the features with ranks of their importance score greater than the data-dependent threshold $\Delta^*$.

$\square$

### Lemma 3: Sharp tail bound for weighted non-central $\chi^2$ distribution

Consider a weighted non-central $\chi^2$ distributed random variable $Y = \sum_{i=1}^{k} u_i Z_i^2$, $Z_i \sim N(\mu_i, 1)$ independently and $\sum_{i=1}^{k} \mu_i^2 = \lambda$. Then for the centralized random variable $X = Y - \sum_{i=1}^{k} u_i(1 + \mu_i^2)$, the following sharp tail bound holds: there exists constants c,c',C>0, such that

$$P(X \geq x) \leq \ c\ exp(-\frac{ct^2}{2k + 2\sum_{i=1}^{k} u_i\mu_i^2 - \sum_{i=1}^{k} u_i}), \forall 0 \leq x \leq c'(2k + 2\sum_{i=1}^{k} u_i\mu_i^2 - \sum_{i=1}^{k} u_i)$$

**Proof**:

The moment-generating function of X:

$$\phi_X(t) = exp\Big[\frac{2t^2}{1-2t}\sum_{i=1}^{k}u_i\mu_i^2 - \frac{k}{2}(log(1-2t)+2t) + t(k - \sum_{i=1}^{k}u_i)\Big]$$

Note, for $0 \le t \le 1/2$, $2t^2 \le -log(1-2t) - 2t \le \frac{2t^2}{1-2t}$.

This implies, for $0 \le t \le 2/5$,

$$t^2(2k + 2\sum_{i=1}^{k}u_i\mu_i^2 - \sum_{i=1}^{k}u_i) \le log(\phi_X(t)) \le 5t^2(2k + 2\sum_{i=1}^{k}u_i\mu_i^2 - \sum_{i=1}^{k}u_i)$$

Next, the exact tail bound can be obtained by applying theorem 1 from Zhang & Zhou (2020).

# B  Appendix 2. Additional Technical Details

In this section, additional technical and implementation details on the proposed algorithm SciDNet are provided.

## B.1  Hyperparameter Selection

Recently developed Deep Learning (DL) models are generally governed by several hyperparameters and properly tuning them is necessary to get effective results. The proposed SciDNet relies on the following hyperparameters: (1) size of the active set $\hat{S}_n$, (2) the intracluster correlation bound $r$, (3) LassoNet tuning parameters $\lambda$ and $M$ and (4) $\kappa$ used in neighbourhood selection in cleaning step. SciDNet is fairly robust to most of the associated hyperparameters. We discuss a practical way to tune all these hyperparameters here:

1. To choose the size of the active set, we propose to select a bigger active set with size proportional to $\nu_n = [n/log(n)]$. As we further cluster the active variables in the clustering step, a slightly bigger active set with boost up the confidence of sure screening property, See the section E for an example.

2. After clustering, the intra-cluster correlation bound $r$ should be fixed at some higher value (usually at 0.9 or 0.95) otherwise the cluster sizes will be inflated.

3. In the cleaning step, a thorough grid search has been done over $\lambda$ considering $\lambda_1 \le \lambda_2 \le \cdots \le \lambda_r$; in practice, a small value is fixed for $\lambda_1$ where all the variables are present in the model. Then the value of the tuning parameter gradually increased up to $\lambda_r$, where there are no variables present in the model. The other hyperparameter for LassoNet is the hierarchy coefficient $M$ for which we follow the path considered in Lemhadri et al. (2021) and set $M = 10$. However, a more flexible approach would be a parallel grid search for $M$ as well.

4. The neighborhood length $\kappa$ can be chosen using the phase transition in the ranks of the importance scores, as described in Section 2.3.

## B.2  Phase transition observed for the CCLE data

The main reason for the phase transition is that, for a null predictor $X_j, j \in S_0^c$, different bootstrap replicates reshuffle its feature importance each time, whereas for a nonnull predictor $X_j, j \in S_0$, the feature importance is much stable in different bootstrap replicates. SciDNet effectively captures this characteristic to identify the null features. As a demonstration, here we present in Figure 10, the bootstrap distribution of rank of the importance scores for the top 25 important cluster representatives via box plots. The green and purple colors respectively indicate if the cluster representatives are selected or rejected by the SciDNet. We can observe the phase transition consistently for all five drugs, and SciDNet selects only those important representatives with reduced variability over the bootstrap replicates.

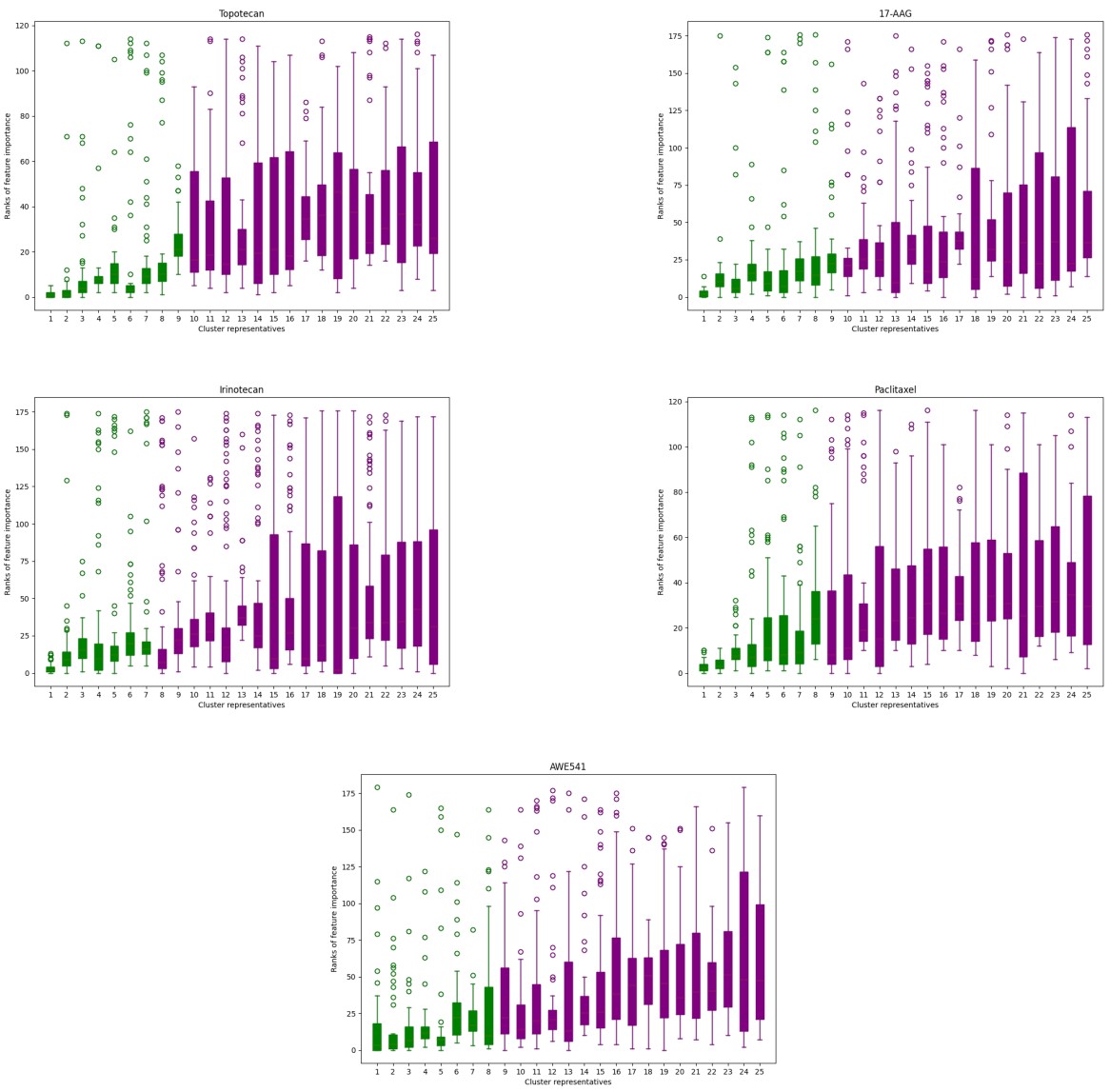

Figure 10: The phase transition property illustrated for the five anticancer drugs considered: (1) Topotecan, (2) 17-AAG, (3) Irinotecan, (4) Paclitaxel, and (5) AWE541, respectively (from top left)

Table S3: Empirical power and observed FDR of SciDNet with standard error in parentheses for gaussian features

| $\rho$ | $snr$ | $q$ | Nonlinear Additive | | | | Nonlinear with interaction | | | | Linear | | | |
|---|---|---|---|---|---|---|---|---|---|---|---|---|---|---|
| | | | 0.01 | 0.05 | 0.1 | 0.15 | 0.01 | 0.05 | 0.1 | 0.15 | 0.01 | 0.05 | 0.1 | 0.15 |
| $\rho = 0.9$ | $snr = 9:1$ | Power | 0.79 (0.19) | 0.93 (0.11) | 0.96 (0.09) | 0.96 (0.09) | 0.99 (0.06) | 1.00 (0.00) | 1.00 (0.00) | 1.00 (0.00) | 1.00 (0.00) | 1.00 (0.00) | 1.00 (0.00) | 1.00 (0.00) |
| | | FDR | 0.00 (0.00) | 0.00 (0.02) | 0.01 (0.03) | 0.02 (0.05) | 0.00 (0.00) | 0.02 (0.05) | 0.03 (0.06) | 0.04 (0.08) | 0.00 (0.00) | 0.01 (0.04) | 0.01 (0.05) | 0.01 (0.05) |
| | $snr = 8:2$ | Power | 0.59 (0.20) | 0.82 (0.15) | 0.86 (0.14) | 0.87 (0.13) | 0.84 (0.24) | 1.00 (0.03) | 1.00 (0.03) | 1.00 (0.03) | 1.00 (0.00) | 1.00 (0.00) | 1.00 (0.00) | 1.00 (0.00) |
| | | FDR | 0.00 (0.03) | 0.00 (0.03) | 0.03 (0.07) | 0.04 (0.08) | 0.00 (0.00) | 0.01 (0.03) | 0.02 (0.06) | 0.04 (0.07) | 0.01 (0.05) | 0.02 (0.06) | 0.02 (0.06) | 0.02 (0.06) |
| | $snr = 7:3$ | Power | 0.42 (0.20) | 0.65 (0.12) | 0.77 (0.14) | 0.81 (0.14) | 0.72 (0.26) | 0.94 (0.12) | 0.95 (0.12) | 0.96 (0.11) | 1.00 (0.00) | 1.00 (0.00) | 1.00 (0.00) | 1.00 (0.00) |
| | | FDR | 0.00 (0.00) | 0.00 (0.00) | 0.00 (0.03) | 0.01 (0.05) | 0.01 (0.04) | 0.03 (0.06) | 0.03 (0.07) | 0.05 (0.09) | 0.00 (0.02) | 0.02 (0.05) | 0.02 (0.05) | 0.02 (0.06) |
| $\rho = 0.95$ | $snr = 9:1$ | Power | 0.83 (0.18) | 0.96 (0.08) | 0.98 (0.06) | 0.98 (0.06) | 0.99 (0.04) | 1.00 (0.00) | 1.00 (0.00) | 1.00 (0.00) | 1.00 (0.00) | 1.00 (0.00) | 1.00 (0.00) | 1.00 (0.00) |
| | | FDR | 0.00 (0.03) | 0.02 (0.05) | 0.03 (0.07) | 0.04 (0.07) | 0.00 (0.00) | 0.04 (0.07) | 0.08 (0.09) | 0.11 (0.10) | 0.00 (0.02) | 0.06 (0.09) | 0.08 (0.10) | 0.11 (0.12) |
| | $snr = 8:2$ | Power | 0.60 (0.29) | 0.82 (0.17) | 0.87 (0.14) | 0.89 (0.12) | 0.98 (0.08) | 0.99 (0.05) | 0.99 (0.05) | 0.99 (0.05) | 1.00 (0.00) | 1.00 (0.00) | 1.00 (0.00) | 1.00 (0.00) |
| | | FDR | 0.00 (0.03) | 0.02 (0.05) | 0.02 (0.06) | 0.03 (0.07) | 0.01 (0.04) | 0.04 (0.07) | 0.08 (0.12) | 0.12 (0.12) | 0.01 (0.03) | 0.05 (0.09) | 0.07 (0.10) | 0.09 (0.11) |
| | $snr = 7:3$ | Power | 0.32 (0.22) | 0.61 (0.19) | 0.79 (0.15) | 0.82 (0.15) | 0.80 (0.27) | 0.96 (0.11) | 0.98 (0.05) | 0.98 (0.05) | 0.93 (0.19) | 0.96 (0.11) | 0.97 (0.09) | 0.97 (0.09) |
| | | FDR | 0.00 (0.00) | 0.01 (0.04) | 0.01 (0.05) | 0.03 (0.08) | 0.00 (0.02) | 0.04 (0.07) | 0.07 (0.09) | 0.12 (0.10) | 0.00 (0.03) | 0.04 (0.08) | 0.06 (0.08) | 0.07 (0.09) |

## C  More Simulation Results

Here we demonstrate finite sample performance of SciDNet under various linear and nonlinear models with varying multicollinearity level under different signal-to-noise-ratio.

### C.1  Using Gaussian Features

For the high dimensional predictors, n i.i.d. copies are first generated from $X \sim N_p(0, \Sigma)$, where $n = 600$, $p = 5000$ and the covariance matric $\Sigma$ is chosen as a toeplitz matrix with $\Sigma_{ij} = \rho^{|i-j|}$. The value of $\rho$ is varied to explore different correlation strength. We set the set of truly significant variables $S = \{100, 200, 300, 400, 500\}$ with $s = 5$. The response $y$ is generated from $y = g(x) + \epsilon$. Here we entertain the following three models:

1. **Linear**: $g(x) = x_S \beta_S$ with $\beta_S$ generated from $N(2, 0.1)$ independently and $\beta_{S^c} = 0$,

2. **Nonlinear additive**: $g(x) = 2x_{100} + 2x_{200}^3 + e^{x_{300}} + 6 \sin x_{400} + 2ReLu(x_{500}^3)$, where ReLu(x)=max(x,0)

3. **Nonlinear with interaction**: $g(x) = 2x_{100} + 2x_{200}^3 + e^{x_{300}} + 6x_{400}x_{500}$

In each cases, the random noise $\epsilon$ is independently generated from $N(0, \sigma^2)$, where the value of $\sigma^2$ is chosen maintaining the signal-to-noise ratio at the desired level. To this end, we define the signal-to-noise ratio as $snr = \frac{var(g(x))}{\sigma^2}$. Here we consider three levels of $snr = 9:1, 8:2$ and $7:3$. Table S3 shows that SciDNet continues to maintain satisfactory power while successfully controlling the FDR below the threshold $q = 0.01, 0.05, 0.1, 0.15$. The average cluster size is observed at 8.3 for $\rho = 0.9$ and 13.4 for $\rho = 0.95$.

### C.2  Using Non-gaussian Features

To check SciDNet's performance under non-gaussian setup, n iid copies of high-dimensional feature vector X are generated from multivariate $t_p(5)$ distribution considering same correlation structure as in the previous section C.1, with n=600, p=5000. The remaining simulation setting is consistent with the previous section 2.1. The performance of SciDNet is presented at table S4 which is quite analogous to the results of gaussian features.

## D  Performance of existing feature selection methods in the presence of high multicollinearity

In this section, we present a numerical illustration of the performance of several recently proposed nonlinear FDR-controlled feature selection algorithms. The predictors are first generated from $X_i \sim N_p(0, \Sigma), i = 1, 2, \ldots, n$, for multiple combination of $(n, p)$ and the covariance matric $\Sigma$ is chosen as a toeplitz matrix with $\Sigma_{ij} = \rho^{|i-j|}, \rho = 0.1, 0.5,$ and $0.9$. Under simplistic setting, the response $y$ is generated from $y = x_S \beta_S + \epsilon$, $S = \{5, 10, \ldots, 50\}, |S| = 10$, with $\beta_S$ generated from $N(\beta_0, 0.1)$ independently and $\beta_{S^c} = 0$. The random noise

Table S4: Empirical power and observed FDR of SciDNet with standard error in parentheses for non-gaussian features

| ρ | snr | q | Nonlinear Additive | | | | Nonlinear with interaction | | | | Linear | | | |
|---|---|---|---|---|---|---|---|---|---|---|---|---|---|---|
| | | | 0.01 | 0.05 | 0.1 | 0.15 | 0.01 | 0.05 | 0.1 | 0.15 | 0.01 | 0.05 | 0.1 | 0.15 |
| ρ = 0.9 | snr = 9 : 1 | Power | 0.68 (0.12) | 0.96 (0.15) | 1.00 (0.09) | 1.00 (0.07) | 0.92 (0.05) | 0.95 (0.02) | 1.00 (0.00) | 1.00 (0.00) | 1.00 (0.00) | 1.00 (0.00) | 1.00 (0.00) | 1.00 (0.00) |
| | | FDR | 0.00 (0.00) | 0.00 (0.01) | 0.04 (0.01) | 0.07 (0.05) | 0.00 (0.00) | 0.01 (0.06) | 0.04 (0.06) | 0.06 (0.07) | 0.00 (0.00) | 0.01 (0.04) | 0.01 (0.05) | 0.01 (0.05) |
| | snr = 8 : 2 | Power | 0.56 (0.24) | 0.86 (0.11) | 0.94 (0.14) | 0.95 (0.13) | 0.76 (0.23) | .93 (0.02) | 1.00 (0.04) | 1.00 (0.03) | 1.00 (0.00) | 1.00 (0.00) | 1.00 (0.00) | 1.00 (0.00) |
| | | FDR | 0.00 (0.01) | 0.00 (0.01) | 0.06 (0.05) | 0.09 (0.07) | 0.00 (0.00) | 0.00 (0.02) | 0.04 (0.03) | 0.07 (0.07) | 0.01 (0.05) | 0.02 (0.06) | 0.02 (0.06) | 0.02 (0.06) |
| | snr = 7 : 3 | Power | 0.42 (0.21) | 0.77 (0.13) | 0.91 (0.16) | 0.94 (0.12) | 0.73 (0.26) | 0.94 (0.12) | 0.95 (0.15) | 0.96 (0.14) | 1.00 (0.00) | 1.00 (0.00) | 1.00 (0.00) | 1.00 (0.00) |
| | | FDR | 0.00 (0.00) | 0.00 (0.01) | 0.02 (0.03) | 0.06 (0.04) | 0.01 (0.05) | 0.04 (0.06) | 0.05 (0.07) | 0.05 (0.09) | 0.00 (0.02) | 0.02 (0.05) | 0.02 (0.05) | 0.02 (0.06) |
| ρ = 0.95 | snr = 9 : 1 | Power | 0.81 (0.19) | 0.95 (0.07) | 0.98 (0.06) | 0.98 (0.07) | 0.99 (0.04) | 0.99 (0.03) | 1.00 (0.00) | 1.00 (0.00) | 1.00 (0.00) | 1.00 (0.00) | 1.00 (0.00) | 1.00 (0.00) |
| | | FDR | 0.00 (0.01) | 0.03 (0.06) | 0.03 (0.04) | 0.05 (0.03) | 0.00 (0.00) | 0.04 (0.07) | 0.08 (0.09) | 0.09 (0.13) | 0.00 (0.01) | 0.03 (0.05) | 0.07 (0.11) | 0.10 (0.14) |
| | snr = 8 : 2 | Power | 0.65 (0.29) | 0.84 (0.16) | 0.89 (0.17) | 0.89 (0.12) | 0.94 (0.07) | 0.97 (0.04) | 0.99 (0.07) | 0.99 (0.07) | 1.00 (0.00) | 1.00 (0.00) | 1.00 (0.00) | 1.00 (0.00) |
| | | FDR | 0.00 (0.03) | 0.01 (0.05) | 0.04 (0.06) | 0.05 (0.06) | 0.01 (0.03) | 0.04 (0.04) | 0.07 (0.14) | 0.11 (0.11) | 0.01 (0.02) | 0.05 (0.09) | 0.06 (0.14) | 0.09 (0.11) |
| | snr = 7 : 3 | Power | 0.47 (0.22) | 0.64 (0.17) | 0.75 (0.19) | 0.87 (0.11) | 0.82 (0.27) | 0.95 (0.10) | 0.98 (0.04) | 0.98 (0.02) | 0.95 (0.10) | 0.96 (0.11) | 0.97 (0.08) | 0.97 (0.06) |
| | | FDR | 0.00 (0.00) | 0.02 (0.04) | 0.02 (0.04) | 0.04 (0.09) | 0.00 (0.01) | 0.04 (0.05) | 0.09 (0.03) | 0.13 (0.09) | 0.00 (0.02) | 0.04 (0.05) | 0.05 (0.08) | 0.08 (0.07) |

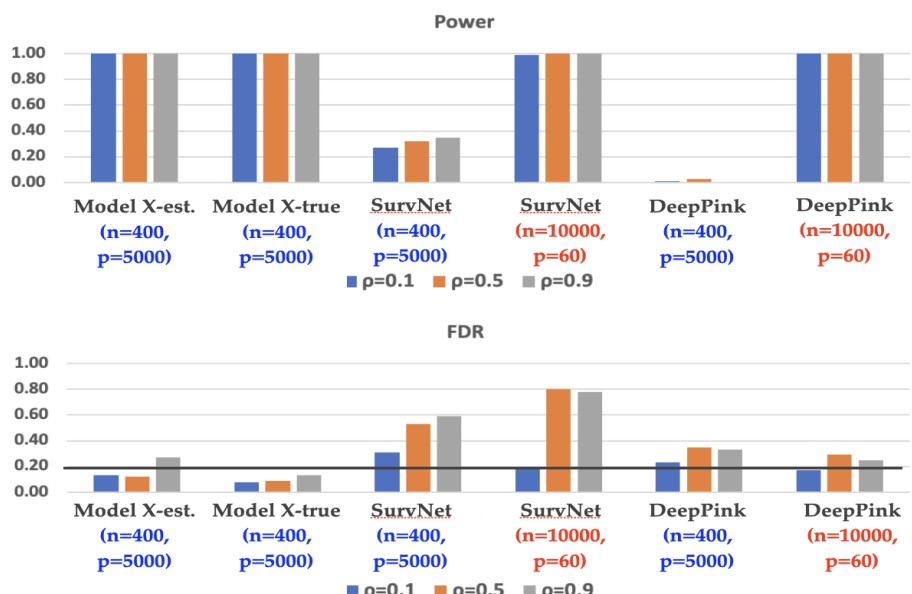

Figure 11: Power and FDR comparison under varying correlation strength and different $(n, p)$ ratio

$\epsilon \sim N(0, 1)$. We focus on the Model-X knockoff (Candès et al., 2016), SurvNet (Song & Li, 2021), DeepPINK (Lu et al., 2018). For a more rigorous analysis, we consider two different versions of Model-X knockoff - (1) Model-X-Estimated, where the knockoffs are generated using an estimated multivariate gaussian distribution and (2) Model-X-True, where the knockoffs are generated using the true data generating multivariate gaussian distribution mentioned above. For the knockoff generation, we consider the equicorrelated construction using the R package knockoff: The Knockoff Filter for Controlled Variable Selection. To implement the SurvNet and DeepPINK, we use the codes mentioned in the respective papers Song & Li (2021); Lu et al. (2018). We set $q = 0.20$ as the FDR control threshold.

Figure 11 reveals several interesting characteristics. Both Model-X-Estimated and Model-X-True maintain the power-FDR balance under a low correlation setup. However with higher multicollinearity, Model-X-Estimated fails to control the FDR below the specified threshold while the Model-X-True controls the FDR efficiently. This disparity indicates Model-X procedure induces inflation in false discoveries if the knockoffs are not generated properly under a 'difficult' situation. As expected, the DL-based algorithms, such as SurvNet and DeepPINK work much better in big-n-small-p and low correlation setups but typically fail in other cases, indicating their reduced effectiveness in ultrahigh dimensional data with small sample sizes.

Table S5: Drug-sensitive genes identified by SciDNet and confirming references

| Drug | Selected clusters of genes | Confirming references |
|---|---|---|
| Topotecan | ***{SLFN11}**,{TUFT1,THRB},{CDT1,SF3A2,SNRPA},{FTH1P10,FTH1},{RPL18},{KLF5},*
*{RPL11,RPL5P4,RPS8,RPL5,RPL10A,AL162151.3,RPS9,RPL3},*
*{KIF15,CCNA2,LMNB1,KIF22,AC009133.14},{MATN2,HSPB8},* | Barretina et al. (2012)
Li et al. (2012) |
| 17-AAG | *{**NQO1**,CTD.2033A16.1},{BAX},{SLC16A3},{PHPT1,SH3BP1}*
*{SPCS3,DCTD},{CTD.2008A1.2,SORD},{NSMCE4A},{CSK}* | Hadley & Hendricks (2014)
Barretina et al. (2012) |
| Irinotecan | ***{SLFN11}**,{KIF15,LMNB1,ARHGAP19},{TCEANC2},{KIF21B},{SQSTM1},{HDAC11}*
*{KHDRBS1,HNRNPA1P35,HMGB2,HNRNPA1,HNRNPA1L2,*
*HNRNPA1P48,HNRNPA1P7,AC021224.1,HNRNPA1P10,RBMX}* | Barretina et al. (2012)
Li et al. (2012) |
| PaclitaXel | *{PARP1,**BCL2L1**},{MMP24},{DIMT1},{RP11.872D17.4,SSRP1,MTA2},{DCUN1D3}*
*{RPL10AP6,RPL10A,EEF2,RPL3},{ARHGAP11B,ARHGAP11A,BUB1B,CASC5},{HCLS1,LCP1}* | Dorman et al. (2016)
Lee et al. (2016) |
| AEW541 | *{TCEAL4,**MID2**},{E2F6},{AC096772.6},{SLC44A1},{PGM1}*
*{ATP8B2,RNF122}, {RP11.1017G21.4},{ETNK2},{NHS},{ATG13}* | Liang et al. (2018) |

## E   Model implementation details and Sensitivity Analysis

In this section, we mention the implementation details of SciDNet that we consider for the simulation study and real data analysis. To select the size of the active set $\hat{S}_n$ in the screening step, in consistence with Xue & Liang (2017), we set $|\hat{S}_n| = \left[\frac{2n}{log(n)}\right]$ by selecting the predictors with the top $|\hat{S}_n|$ Henze–Zirkler test statistic $\tilde{w}_k^*$ , where $[z]$ denotes the integer part of z. In all our simulation scenarios, we set r=0.9, the hyperparameter for intra-cluster correlation bound to further integrate highly correlated conditionally dependent clusters. In cleaning step, for LassoNet 100 dimensional one-hidden-layer feed-forward neural network has been used; more detailed model architecture can be found at appendix in Lemhadri et al. (2021). For creating the compact neighbourhood in the cleaning step, each time we choose the value of $\kappa$ utilizing the phase transition property mentioned in section 2.2 of main manuscript. The feature selection performance of the SciDNet is demonstrated by calculating the average power and cFDR along with their standard error observed in 50 Monte Carlo replications. Each data set is randomly divided into train, validation and test with a 70-10-20 split. To asses the prediction performance, the test Mean Square Error (MSE) before and after the variable selection has been shown as part of the simulation study. For the prediction model, a 40-dimensional two-hidden-layer feed-forward neural network with ReLU and linear activation function is considered with Adam as optimizer. For the regression tree, we used the bagging for further stabilization, as mentioned in Breiman (1996). The number of leaves and nodes are chosen by minimising the MSE on validation set.

To access the error bar for the sensitivity analysis, we generate typical data using the polynomial setup (section 3 in the main manuscript, with $\beta = 2, \sigma^2 = 1$) and rerun SciDNet 50 times on the same data and set $q = 0.1$ as FDR-control threshold. The mean and standard deviations from these 50 replications are following: **power** $= 0.99$ $(0.01)$, **observed FDR** $= 0.03$ $(0.03)$, **test error by LassoNet** $= 1.048$ $(0.101)$, **test error by SciDNet+RT** $= 0.710$ $(0.002)$.

## F   Important clusters of gene discovered by SciDNet

### F.1   For CCLE dataset

Following Table S5 presents all the selected cluster of genes by SciDNet for the five anticancer drugs considered. The genes in a single cluster are mentioned in the "{}". Previous research on this gene-expression data has revealed several genes as biologically associated with the corresponding drugs. SciDNet successfully discovers these genes as the top-most important gene associated with the drugs.In Table S5, the selected genes which are confirmed by previous domain research, are highlighted and corresponding references are mentioned in the column 3.

### F.2   For Riboflavin dataset

The Riboflavin production dataset contains much complicated correlation structure than the CCLE data, see Figure 1 in the main manuscript for a visual illustration. As a result, SciDNet has produced much larger

Table S6: Selected clusters of genes by SciDNet applied in the riboflavin gene data example

| Cluster No. | Genes selected |
|---|---|
| 1 | EPR_at, IOLD_at, KAPB_at, PROJ_at, RPLQ_at, UREA_at, YCGB_at, YCGM_at, YCGN_at, YCSN_at, YCGO_at, YCGT_at, YDBM_at, YHXA_at, YKZC_at, YOAB_at, YPJB_at, YUSX_at, YVFH_at |
| 2 | COMX_at, CSPC_at, HAG_at, MPR_at, YBDL_at, YDBM_at, YHCB_at, YJFB_at, YHFS_at, YOAB_at, YODF_at, YOAC_at, YONU_at, YOTL_at, YQKI_at, YQZH_at, YTEI_at, YUSV_at |
| 3 | HIT_at, KATX_at, LICH_at, NASA_at, OPUCB_at, PHRG_i_at, PHRK_at, ROCB_at, ROCR_at, SACB_at, SPOIIE_at, TMRB_at, YACN_at, YBBJ_at, YBGB_at, YCBF_at, YFKJ_at, YHCS_at, YHXA_at, YJBF_at, YLBA_at, YLOU_at, YPUI_at, YQGY_i_at, YUKE_at, YVVYD_at, YXLJ_at, YXZF_at |
| 4 | APPA_at, BGLS_at, ccpB_at, MMR_at, SIGY_at, SOJ_at, TREA_at, YBGB_at, YDGF_at, YOPR_at, YQEB_at, YVCI_at, YVDR_at, YWBG_at, YWDE_at, YWFM_at, YXBB_at, YXIL_at, YXIO_at, YXIQ_at, YXJA_at, YXJN_at, YXLC_at, YXLD_at, YXLE_at, YXLF_at, YXLG_at, YXLJ_at, YXZF_at, YYBF_at |
| 5 | LYTD_at, SQHC_at, XKDE_at, YFIG_at, YFIH_at, YFII_at, YFNC_at, YHDV_at, YIST_at, YJGA_at, YTCP_at, YTMP_at |
| 6 | YCDH_at, YCDI_at, YCEA_at, YCIA_at, YCIB_at, **YCIC_at**, YDAR_at, YHZA_at, YRPE_at, YTGA_at, YTGB_at, YTGC_at, YTGD_at, YTIA_at, YVQH_at |
| 7 | OPUBD_at, PHRE_at, SIPS_at, YBFF_at, YDEM_at, YNAB_i_at, YNAC_at, YNEK_at, YOBF_at, YOKG_at, YONX_at, YOPA_at, YOPR_at, YOTL_at, YPBB_at, YQZH_at, YRDA_at, YRKK_at, YRKL_at, YTGB_at, YUXI_at, YWCE_at, YWQK_at, YYDB_at, YYDF_i_at |
| 8 | ARGB_at, ARGC_at, ARGD_at, ARGJ_at, CARA_at, CARB_at |
| 9 | PROJ_at, RPLF_at, RPLJ_at, RPLL_at, RPSN_at, RPSP_at, YLQC_at |

cluster of genes compared to the cluster sizes from the CCLE dataset. For example, the average cluster size for CCLR and Riboflavin dataset is respectively 2.5 and 17.78. Following Table S6 shows the 9 selected cluster of genes selected by SciDNet while the FDR is controlled at $q = 0.15$. Additionally, SciDNet discovered the gene *YCIC_at* as one of the expressive genes related to riboflavin production which was identified by Bühlmann et al. (2014) as a causal gene in this context.

