# OpenReview forum: "Error Controlled Feature Selection for Ultrahigh Dimensional and Highly Correlated Feature Space Using Deep Learning"
_TMLR — Rejected by TMLR_

### Review · Reviewer_yi1K · 2023-06-27

**Summary Of Contributions:**

The authors propose an approach to identify highly correlated data in ultra-high dimensions based on deep learning. The argument lies within the fact that multidimensional data, mainly those encountered in genomics, exhibit high degrees of colinearity, which begs the question of how one can more effectively deal with this. There have been a few methods proposed that mitigate this, e.g. LassoNet, Knockoffs, etc. but this paper proposes an approach that identifies clusters of variables that share common characteristics and use this cluster as a proxy to any learning methodology. The proposed approach demonstrates good performance compared to other methods on a number of simulated and real datasets and on ablation studies.

**Audience:**

Yes

**Claims And Evidence:**

Yes

**Requested Changes:**

a) Please migrate unnecessary mathematical formulation in the appendix or simplify the flow.
b) Reflect on the practical relevance with respect to 'do the current DL system need to factor collinearities'?
c) Consider adding a baseline DL method in the mix.

**Strengths And Weaknesses:**

The method is simple, well articulated and motivated, but the paper is hard to follow. The first two pages are well written providing a very nice background and building up the momentum well, but the method description and results are presented in an overly complex manner, especially pages 4 and 5. I would suggest simplifying the paper and moving more parts into the appendix.

In addition, the problem is somewhat ill-defined in that it does not stress whether current deep learning systems suffer from the collinearities in high dimensional data with respect to absolute performance for the task at hand. I appreciate that this might not be the main focus of this work and that eliminating information is both computationally relevant and important from an explainability perspective.

---

> ### Author Response · Authors · 2023-07-28
> **Response to Reviewer yi1K**
>
> We sincerely appreciate your comprehensive review and valuable feedback on our manuscript. This opportunity allows us to address the concerns raised and further enhance the clarity of our work.
>
> 1.	In response to the reviewer's concerns about the mathematical parts, we have carefully revised and clarified several notations and definitions in the manuscript. Additionally, we have incorporated a schematic overview of the entire algorithm to facilitate a better understanding of SciDNet's approach. However, we consider the mathematical details for the screening part to be fundamental in providing a comprehensive algorithmic description of SciDNet. Therefore, we have retained these crucial details in the main manuscript.
>
> 2.	As highlighted in Figure 1 and Figure 5 of the main manuscript, current feature selection methods, both in deep learning and knockoff frameworks, are greatly affected by the substantial multicollinearity in the feature space, leading to reduced power and an abundance of false discoveries. We believe these empirical results emphasize the significance of our proposed approach in overcoming such challenges.
>
> 3.	To ensure valid comparisons, we used the LassoNet algorithm as the baseline in our experiments. The LassoNet algorithm possesses sparsity-inducing properties and offers a simplistic yet effective comparison points with our method.

---

### Review · Reviewer_j8Mo · 2023-07-02

**Summary Of Contributions:**

This paper proposes an algorithm for feature selection called SciDNet- Screening & Cleaning Incorporated Deep Neural Network. The algorithm essentially first selects a set of candidate variables using a Henze–Zirkler statistical test with some clustering of the vvariables, then uses a regularized deep neural network to rank the importance of the clusters and select a final set of variables for a given false discovery rate.

The new knowledge presented by this paper is the algorithm as a whole, which as far as I am aware is novel (but I am not at all an expert in this sub area). However the algorithm clearly builds on prior work, and I don't have a clear understanding of how the series of steps proposed in this paper really differ from prior work or trivial extensions of it.

**Audience:**

Yes

**Broader Impact Concerns:**

no concerns

**Claims And Evidence:**

No

**Requested Changes:**

## Clarity

I think this is the most critical thing to improve in the paper. I think the paper was unclear for multiple reasons which intersect and compound, so it is hard to pinpoint an exact cause to the lack of clarity. I have done my best to outline as many reasons as possible for it not being clear, but I think the ultimate metric for clarity is whether the paper is easy to read and helps the reader clearly understand the problem, the solution, and the desirable aspects of the solution.

**Reason 1: unclear math**

I think the problem needs to be formalized more precisely, for example:

- $X=(X_1,\ldots,X_p)$: upon first reading, it is unclear whether this is a set of $p$ observations or one $p$-dimensional observation. In the end I think it was the latter.
- $E(FDR)$ was unclear to me. Is $E$ expectation? That is usually stylized as $\mathbb{E}$. If it is expectation, what is the expectation with respect to?
- Are datapoints assumed to be iid?
- What is $O(...)$? Is it big-O notation? This describes asymptotic behaviour, however it is unclear what asymptotic behaviour is referred to in this work. For example, writing $p=O(exp(n^\tau))$, does that mean the dimension should grow as the dataset size grows, at this particular rate? If you just want to say "high dimensional" either formalize it more precisely or simply state "high dimensional". Similarly, what is $|S_0|=O(1)$ mean? I did not have a guess for this one.

**Reason 2: alluding to rather than explaining previous work**

The authors sometimes use a writing style that refers to previous work without giving much indication as to what that work says. For example:

> Following the intriguing arguments in Rudin (2019), caution must be exercised regarding the application of DL models for
decision-making in real-world problems

>  In the spirit of Liu et al. (2009), we assume that the predictors follow the nonparanormal distribution

If the reader has not read these works, then the point is completely lost. Generally when referring to work in this way it is best if the reader does not need to go to another paper to understand your argument. Papers should be self-contained! I think the authors should aim to include all necessary claims in the paper, referring to other work merely for an _elaboration_ of those points.

**Reason 3: no "overview" of algorithm**

Given all the steps required for the algorithm, I think it would be useful to have an overview of the algorithm beyond the mention of "screening and cleaning" in the introduction. This could either take the form of an algorithm block or a schematic. This is just a suggestion though. However, I think one reason I was not able to feel like I understood the method in the time I allocated to this paper was because I had trouble understanding how all the various parts fit together. It would also be useful to understand how different parts in the proposed pipeline differ from previous work, both in isolation but also considering the algorithm as a whole. For example: how does the selection procedure differ from other selection procedures, and how does the outcome of the selection procedure differ / compare to other algorithms without an explicit selection step?

On this note: it would be good to have an explicit algorithm block for all algorithms in the paper.

## Experiments

Beyond clarifying whether discrete genetic data is covered by the assumptions made by the method, I suggest either doing experiments on other data types or mentioning why you think this method is especially well-suited to genetic data.

Also, I think it is important to compare to more than 1 other method for real-world data if you want to give the reader a good understanding of the empirical performance.

**Strengths And Weaknesses:**

As a preface to my assessments of the the strengths and weaknesses of this paper, I want to clarify that:

1. Feature selection is not my particular area of expertise: I know the basics and the general goal, but am not deeply familiar with the subtleties of this area and important recent developments.
2. Due to other commitments I was only able to spend one afternoon reviewing the paper, and despite my best efforts I still feel like I don't *get* the paper. Therefore some of my comments may be misguided or wrong. However, even if this is the case, the authors should take this as a sign that the clarity of the paper could be improved (more on this below).

*Strengths*

- focusing on a classic / important problem
- Included a good discussion on false discovery rate: it is important to consider the trade-off between accuracy and false discovery in real-world problems

*Weaknesses*

- Writing and rhetoric was unclear. I think it is hard for the reader to get a good overview of the algorithm described and the field in general from reading this paper. More on this in "requested changes" section.
- Despite claiming that SciDNet "relies on minimal modeling assumptions", it actually seems to rely on some quite strong assumptions. It is not clear how realistic these assumptions are and how they limit the utility of this method. In particular, the assumptions I spotted are:
  - That the only depends on a small subset of variables and that the rest are noise, in particular $Y$ is conditionally independent of most inputs given the "important" ones.
  - That $X$ has a nonparanormal distribution, i.e. that it can be realized by sampling from a Gaussian and non-linearly distorting each output dimension independently. This seems very limiting: for example, I think that the classic "2 moons" dataset ([example](https://machinelearningmastery.com/wp-content/uploads/2017/12/Scatter-plot-of-Moons-Test-Classification-Problem-768x576.png)) can be expressed in this way because the number of modes changes depending on the $x_2$ variable, but for the paranormal distribution the non-linear transformation is the same for each dimension. However, I acknowledge this is not a full proof, so I could be wrong.
- Although the paper did mention some prior works in the introduction, I felt there was not much discussion of how the algorithm differed from / improved upon / addressed limitations of these prior works. It felt like it was missing a "related work" section. For somebody unfamiliar with the pros/cons of many methods, such a section would be helpful.
- Experiments were only on genetic data. In principle there is nothing wrong with using only one data type, but given that the method is claimed to be quite general I was somewhat surprised to see this. Furthermore, if I understand correctly genetic data is discrete (specifically a discrete sequence), but the main subject of this paper is the selection of continuously-distributed features. I am confident that no discrete distribution is expressible as a nonparanormal distribution (because it would need to be a continuous + differentiable transformation of a Gaussian distribution, thereby also making it continuous and differentiable). Also, only 1 other method was used in the comparison.
- Length: for what it is, I think the paper could be shorter. A lot of the explanations felt non-critical and the authors could likely shorten it without losing much (in fact, it might even gain clarity)

---

> ### Author Response · Authors · 2023-07-28
> **Response to Reviewer j8Mo**
>
> We extend our gratitude for the comprehensive review and valuable feedback provided on our manuscript. We would like to utilize this opportunity to address the concerns and enhance the clarity of our work.
>
> ### **Assumptions**:
> 1.	The sparsity assumption, where only a small subset of features is truly associated with the response, serves as a fundamental principle in feature selection methods, justifying the pursuit of optimal subset selection. This intuition is grounded in domain-specific knowledge as well. For instance, in genetics, it is often expected that only a few genes are associated with a phenotype y of interest.
> 2.	As an alternative to the knockoff-based procedure, which requires exact knowledge of the feature's distribution, we employ the non-paranormal assumption as a middle ground between fully specifying the relationship between the response y and predictors X (as in traditional feature selection methods like lasso and elastic net) and quantifying the feature distribution (as in the knockoff framework). The gaussian-copula based nonparanormal family allows for flexible modeling, encompassing a wide range of parametric distributions while preserving the conditional dependency structure in the feature space. Notably, the use of nonparanormal distribution as a semi-parametric model for feature distribution remains non-trivial and novel in the existing feature selection literature. In our future research, we plan to explore generative algorithms to address more complex examples, which would potentially include extensions of the two-moon dataset.
>
> ### **Related works**:
>
> To enhance clarity, we have included a dedicated section in the introduction for the ease of demonstration. This section is divided into two main subsections: (1) FDR controlled feature selection and (2) Addressing uncertainty in highly correlated feature space. While feature selection has been a prominent area of statistical research for the past few decades, we have focused our discussion on recent developments in model-free feature selection to keep the discussion concise and relevant. The revised manuscript now includes a titled section in the literature review to provide readers with a clearer understanding of the content.
>
> ### **Clarity**:
>
> 1.	Throughout the paper, we use the following notation: the feature space is p-dimensional, denoted as $X={X_1, X_2, \dots, X_p}$, and we observe $n$ independent and identically distributed (iid) data points, with each data point represented as $(Y_i, X_i)=(Y_i, X_{1i}, X_{2i}, \dots, X_{pi})$, where $i$ refers to the data point index.
>
> 2.	Unless specified otherwise, the probability measure $P(\cdot)$ or the expectation $E(\cdot)$ is taken with respect to the overall data-generating randomness, which includes both $(Y,X)$.
>
> 3.	We have included an explanation of the "big-O" notation in the current context, clarifying its meaning and usage for the readers' better understanding.
>
> 4.	To provide an overview of the entire algorithm, we have added a schematic figure in Section 1 of the revised manuscript.
>
> The proposed approach holds significant applicability in genetics and neuroimaging studies, where datasets often consist of ultra-high dimensional features (SNPs or voxels) with high inter-feature correlations, posing challenges for traditional feature selection methods. Notably, in these domains, each feature carries a meaningful physical interpretation, such as a specific genetic expression level or DMRI metrics representing human brain white matter tracts, making feature selection even more compelling. Hence, to demonstrate the efficacy of our method, we utilize two publicly available gene-expression datasets. However, it is essential to emphasize that SciDNet's applicability extends beyond genetic or neuroimaging studies, finding relevance in various other domains as well.
>
>
> We express our sincere gratitude to the reviewer for providing thoughtful and constructive comments. The suggested edits have undoubtedly enhanced the comprehensiveness of the manuscript.

---

> > ### Comment · Reviewer_j8Mo · 2023-08-11
> > **Response to revisions**
> >
> > I have looked at the revised paper (sorry for taking so long to do so). FYI, in ML venues authors often put text changes in red during the revision to make it obvious to reviewers what parts of the paper have changed. Reviewers are busy and combing through a 15 page PDF to check what has changed is not a great use of time, especially if it is a paper you are not very excited about (no offence intended). It appears that the changes made were fairly minimal.
> >
> > Thank you for making changes though. My thoughts on these changes are:
> >
> > - **Clarity:** I think the changes have improved the clarity, but since the changes were minimal the improvements are also minimal. My main concern was that it is hard to get an overall sense of the algorithm. While the schematic in the introduction is certainly helpful, it would not be sufficient for somebody to re-implement your method. Given that the code provided is not really well-documented, I think including an explicit algorithm block might help.
> > - **Related work:** it was not immediately obvious to me what was added here when I compared with the previous version of the paper. It seems the changes are minimal and I still think the paper could benefit from a better overview.
> > - **Assumptions:** at TMLR one of the main reviewing criteria for papers is "are the claims correct." One of the claims you make in section 1 is that your method "relies on minimal modelling assumptions" unlike other methods. This is inherently a subjective claim (i.e. what is "minimal"), and my subjective assessment was that I disagreed with the claim. I agree with what you have written in your response that these assumptions have merits and am fully aware that it is not possible to do learning without assumptions, but ultimately I find it hard to describe assumptions as "minimal" when they are violated by common toy datasets in the field like "two moons". My current opinion is therefore that the claim is not correct. I hoped the authors would add more nuance to the claim in the revision.
> >
> > Furthermore, there are some questions from my review which you did not answer:
> >
> > - The experiments are with discrete genetic data, but the method is designed for continuously-distributed data.
> > - Big O notation: I had the impression the authors were trying to formalize the notion of "very high dimension" using big-O notation when this didn't really make sense. The change made by the authors essentially just describes the big O notation in words, which doesn't really address the core issue that I don't think there is a well-defined limit which would justify using big O. The exact form doesn't seem to be used anywhere: why is it $O(exp(n^\tau)$ and not $O(exp(n!))$ or $O(exp(n^{n!}))$? These are all "really big numbers". Why is one limit correct?
> >
> > I would appreciate your response to my concerns, since at the moment I am uncertain whether I should recommend acceptance.
> >
> > Also, on a more personal note, please don't feel the need to grovel to me in your responses (e.g. "sincere gratitude", "undoubtedly enhanced", etc). Maybe some people like this but I personally find it a little bit weird 😳

---

> > > ### Author Response · Authors · 2023-08-17
> > >
> > > Thanks for the comprehensive review of our paper. Please find below our responses to the raised concerns:
> > >
> > > * **Clarity**: The proposed SciDNet method is designed to effectively tackle the uncertainty stemming from extreme multicollinearity within feature spaces during the feature selection process. It consists of two key components: Screening (integrating dimension reduction and clustering) and Cleaning (ensuring FDR control).
> > >     1. The sure screening property guarantees that crucial features are retained after dimension reduction in the screening stage.
> > >     2. Leveraging conditional dependencies aids in clustering highly correlated features within the same group.
> > >     3. Ultimately, non-linear FDR estimation facilitates statistical inference, yielding a concise selection of features with controlled FDR at a user-defined level.
> > >
> > > We have summarized these crucial points in the contribution section and supported them with an accompanying schematic figure. If deemed helpful, we are prepared to include a more comprehensive algorithmic workflow in the appendix as supplementary material.
> > >
> > > * **Related work**: While feature selection is a significant research area within statistical ML, there remains a limited body of work addressing the intricate uncertainty associated with highly correlated features in a nonlinear, nonparametric context. In our revised manuscript, we have categorized the section on related works (within the introduction) into two primary classes:
> > >     1.	Error-controlled feature selection: We introduced the concept of FDR and delved into conventional model-based feature selection methods, along with recent advancements in knockoff frameworks highlighting the challenges we seek to address in this study.
> > >     2.	Uncertainty in highly correlated feature spaces: In this subsection, we specifically addressed the issue of extreme multicollinearity, a common obstacle in traditional feature selection techniques. We examined alternative approaches in the literature, including p-value-based model-driven methods and those from the knockoff family that requires the estimation of high-dimensional feature distributions. We underscored the limitations of both these approaches using empirical evidence (Figures 1 and 2) before transitioning to our contribution section to introduce our proposed method.
> > >
> > > While we acknowledge that the realm of feature selection boasts a wide array of algorithms, we primarily centered our attention on recent techniques within the nonparametric nonlinear domain, both for the literature review and our empirical evaluation. Should the reviewer have any prior works in mind that are pertinent to our proposed methodology, we welcome the opportunity to consider and assess those options.
> > >
> > > * **Assumptions**: We categorize existing approaches into two main groups: (1) model-based, which assume parametric relationships between features and the response for parameter inference, and (2) knockoff-based methods, which don't assume any relationship but rely on precise prior knowledge/estimation of feature distributions to generate knockoff features. In our work, we adopt the term 'minimal assumption' to convey:
> > >     1. No need to specify a specific parametric relationship between the response and features; we use DNN to approximate any nonparametric function.
> > >     2. No requirement to specify/estimate feature distributions, which can be challenging; instead, we assume a nonparanormal family, encompassing various parametric distributions.
> > >
> > > From our experience, the nonparanormal transformation performs well in practical scenarios. We applied this transformation to the **"two-moon dataset"** mentioned by the reviewer and observed that it preserves the dataset's inherent shape while converting it into a Gaussian distribution (evident in the QQ-plot's near-linear trend for the transformed variable, [an illustration can be found here](https://drive.google.com/file/d/1-wF0p8xwepoNuMLLvfn1WD71cmwQOBvp/view?usp=sharing)). Theoretically, this is justified by the nonparanormal family's capability to incorporate distinct unknown functions for each dimension, connecting them to a Gaussian distribution. Upon the reviewer's request, we can add an appendix section to assess the flexibility and robustness of the nonparanormal transformation using simulated and synthetic datasets such as "two moons."

---

> > > > ### Author Response · Authors · 2023-08-17
> > > >
> > > > Please find below our responses to your other comments.
> > > > 1. **Continuous gene-expression data**: The genetic data we analyzed is continuous gene-expression data, while the phenotypes we used as responses are also continuous variables. Nevertheless, our approach can readily be extended to encompass discrete variables by employing apt screening methods and adapting the loss function within the DNN framework, as discussed in the conclusion section.
> > > > 2. **Big-O notation**: While we acknowledge the reviewer's observation that there could be various ways to express "ultra-high dimensionality," our selection of exponential growth is rooted in the existing literature (e.g. [Zhao & Yu, 2006](https://www.jmlr.org/papers/volume7/zhao06a/zhao06a.pdf?ref=https://githubhelp.com), [Fan and Lv, 2008](https://academic.oup.com/jrsssb/article/70/5/849/7109492)). Moreover, this choice embodies the practical concept of the rapid growth of 'p' in relation to 'n,' aiding in establishing concentration bounds within theoretical analyses in ML literature.

---

> > > > > ### Comment · Reviewer_j8Mo · 2023-08-28
> > > > > **Thanks for clarifications**
> > > > >
> > > > > Sorry for the delayed response to your most recent messages.
> > > > >
> > > > > - I think an overall algorithm statement in the appendix would be useful
> > > > > - I think related work description is a bit better. I don't have any particular works in mind for you to cite.
> > > > > - Assumptions: I agree that the two-moon dataset can be approximated by a non-paranormal distribution as you show in your figure, but strictly speaking it cannot be described with a non-paranormal distribution, no? Data which exists in several scattered clusters that aren't axis aligned also seems like it would generally not be describable by a non-paranormal distribution. I don't think including the two-moons diagram in the paper is useful, but I think a more nuanced statement of the implications of this assumption would be nice.
> > > > > - Gene data: I didn't know it was continuous, maybe explain the data more clearly?
> > > > > - Big O notation: if it is in previous papers then fair enough I guess, but it still seems imprecise to me to use big O notation when not referring to explicit asymptotic behaviour.

---

### Review · Reviewer_WtgE · 2023-07-18

**Summary Of Contributions:**

Overall, the paper is clear and well-written. This paper proposes a novel method that combines deep learning with screening and cleaning techniques to control the false discovery rate (FDR) in feature selection for black-box model interpretation purposes. This work falls within the emerging and novel field of knockoff statistics which is a powerful framework to control FDR with theoretical guarantees. This paper is novel in several aspects compared with existing literature







**Audience:**

Yes

**Broader Impact Concerns:**

No concerns.

**Claims And Evidence:**

Yes

**Requested Changes:**

This paper misses some key papers in the knockoff literature in their discussion. For example, the authors mentioned that "we adopt the cluster version of the FDR as the expected value of the "proportion of clusters that are falsely declared among all declared clusters". This is very related to the notion of group knockoff statistics for FDR control. These main papers in the context of group knockoffs are not cited:
a. A Prototype Knockoff Filter for Group Selection with FDR Control https://arxiv.org/pdf/1706.03400.pdf b. The knockoff filter for FDR control in group-sparse and multitask regression at https://proceedings.mlr.press/v48/daia16.html c. Deep-gKnock: Nonlinear group-feature selection with deep neural networks.

The authors should also briefly discuss the relationship of their paper to these aforementioned papers.

In general, the simulation study is well-designed and discussed. However, some other aspects should also be explored.
a. In Appendix C.1, the equation of wo nonlinear relationship "Nonlinear additive" and "Nonlinear additive with interactions" should be emphasized and mentioned in the main text. The reason is single-index model is still similar to a linear relationship. The nonlinear additive and the one with interactions better support the results for a nonlinear relationship. The authors should emphasize that they have explored more complicated cases other than the single-index model.

b. Since the authors explore the features by clustering them and selecting cluster representatives, to test the effectiveness for this, the authors can design one simulation setting by varying within-group variation with rho ={0.2, 0.4, 0.6, 0.8} and between-group correlation rho={0.2, 0.4, 0.6, 0.8} to see if their cluster method is effective enough to differentiation the group membership of the features. c. For the current existing simulation in Figure 4, instead of correlation rho=(0.1, 0.5, 0.9), more different rho values such as 0.2, 0.3, 0.4, 0.6, 0.7, 0.8 should also be explored to should the trend. Instead of using bar charts, it may be more effective to show performance comparison using multiple curves with different rho values with a fixed signal noise value like 5.

In terms of real data analysis: a. Untangle the response mechanism of anticancer drugs using Knockoffs method has been explored in Zhao, Tingting, Guangyu Zhu, Harsh Vardhan Dubey, and Patrick Flaherty. "Identification of significant gene expression changes in multiple perturbation experiments using knockoffs." Briefings in Bioinformatics 24, no. 2 (2023). This work should also be cited.
b. The authors describe the response variable as "For the response, we used the activity area (Barretina et al.,2012)", a little more information should be provided to describe what type of variable is activity area? Is it continuous or categorical, explain using one sentence about how genes could affect the activity area. Is this a classification task or a regression task?

c. In the last paragraph of page 13, the authors discuss some important genes selected by their method, is the corresponding coefficient positive or negative for these selected genes and how this is related with the upregulation or down-regulation of the genes and how this affects the response variable? More biological interpretations should be provided. Also, is this consistent with existing literature?

Minor issue, Under section 3.1, "we present .....", letter w should be capitalized.

**Strengths And Weaknesses:**

Strengths:

It enables reproducible high-dimensional nonlinear-nonparametric feature selection in the presence of highly correlated predictors.
It performs screening by taking advantage of nonparanormal transformation and HZ test.
Then it clusters the active predictors using the precision matrix, which is described in Algorithm 1.
To finally control FDR, it proposes a novel importance score computation method for each cluster representative using Deep Neural Networks (DNNs) followed by a bootstrap resampling method.

Weakness:

1. The paper misses some very related recent papers in the knockoff literature and does not discuss the relationship with these previous works.

2. The simulation study is well-designed and discussed. However, some additional aspects should also be explored.

3. More intuitive description and biological explanation for the real dataset should be provided.

---

> ### Author Response · Authors · 2023-07-28
> **Response to Reviewer WtgE**
>
> We sincerely appreciate your valuable feedback and positive assessment of our manuscript. We addressed your comments in the following sections.
>
> ### **Literature on Knockoff Frameworks**:
> The knockoff framework relies on prior knowledge of the predictor's distribution to generate knockoff features. However, in ultra-high dimensional settings, estimating the predictor's distribution can be a significant bottleneck, as shown in Figure 1 and confirmed by Barber et al. (2020). To address this, we propose an alternative by assuming a minimal distributional assumption—wherein the features jointly follow a nonparanormal distribution. This choice strikes a balance between model-based feature selection (e.g., lasso, elastic nets) and model-free feature selection (e.g., knockoff frameworks), requiring estimation of the feature's distribution. We discuss these distinctions in the introduction to differentiate our approach from the knockoff framework.
>
> Regarding the references mentioned by the reviewer, we acknowledge their relevance in handling grouping structures in low-dimensional linear regression settings. However, it is important to note that our paper focuses on the ultra-high dimensional model-free feature selection scenario, which presents distinct challenges compared to the mentioned works. Therefore, while these references are valuable contributions, their direct generalization to our current paper's setting may not be straightforward.
>
> ### **Updating the simulation study**:
> We value the reviewer's positive assessment of our simulation study's structure. However, we wish to reiterate that SciDNet, as a data-adaptive method, forms multi-resolutional clusters of features based on their estimated conditional dependency structure. As such, we cannot predefine the intra and inter-group correlation in the simulation study. To address this concern, we have included a new simulation study in Section 3.4. In this study, we utilize a nonlinear additive model as the data-generating mechanism and vary the autocorrelation between features as $\rho=\{0.1, 0.2, …, 0.9\}$. Figure 7 presents the power, FDR, number of selected features, and cluster size. The results demonstrate that SciDNet effectively maintains the power-FDR trade-offs by adaptively adjusting the cluster size and handling the added uncertainty in feature selection in highly correlated feature spaces. We believe this new analysis further solidifies the robustness of our proposed method.
>
> ### **Further Insights into the real data analysis**:
> We applied SciDNet to the publicly-available cancer cell line encyclopedia (CCLE) dataset for identifying genes associated with anticancer drug sensitivities. This dataset contains 8-point dose-response curves for 24 chemical compounds across over 400 cell lines, with each cell line having expression data for 18,926 genes. We used the area under the dose-response curve, known as the activity area in Barretina et al. (2012), as the response variable to measure drug sensitivity. The activity area represents the extent to which a drug enters a person's bloodstream after administration and is a continuous variable.
>
> In this ultra-high dimensional regression setting, our objective was to use SciDNet to identify genes associated with drug sensitivity for each drug. To validate the performance, we selected 5 out of 24 drugs with drug sensitivity genes verified by other studies for testing. The results from SciDNet are consistent with previous studies, and we provide a summary of the selected genes and corresponding confirmatory literature in Table S5 of the supplementary material.
>
> We would like to highlight that although SciDNet captures a more complex non-linear relationship, the non-parametric nature of its deep-learning model makes the individual signal strength less interpretable compared to a linear model with coefficients. Despite this limitation, it is important to note that the genes selected by SciDNet remain relevant to the activity area, as evidenced in Table 1. Notably, even after substantial dimension reduction, SciDNet achieves and maintains high prediction accuracy with only a small number of selected features.
>
> Furthermore, we observed that the features selected by SciDNet are highly correlated with the outcome variable. For instance, SciDNet identifies the gene SLFN11 as strongly associated with the outcome, which aligns with existing literature. Additionally, we calculated the correlation coefficient between the outcome activity area and gene SLFN11, resulting in 0.46, while 95% of the genes have a correlation coefficient of less than 0.27. This finding indicates that the selected genes indeed play a crucial role in understanding the nature of dose-response curves for different drugs.
>
> We appreciate your valuable feedback and have incorporated the suggested information into the revised manuscript, improving its quality. Thank you for your thoughtful suggestions.

---

### Comment · Reviewer_WtgE · 2023-07-15
**Combining Deep Learning with Screening and Cleaning Techniques for Controlling False Discovery Rate in Feature Selection for Black-Box Model Interpretation**

Overall, the paper is clear and well-written. This paper proposes a novel method that combines deep learning with screening and cleaning techniques to control the false discovery rate (FDR) in feature selection for black-box model interpretation purposes. This work falls within the emerging and novel field of knockoff statistics which is a powerful framework to control FDR with theoretical guarantees. This paper is novel in the following aspects:

1. It enables reproducible high-dimensional nonlinear-nonparametric feature selection in the presence of highly correlated predictors.
2. It performs screening by taking advantage of nonparanormal transformation and HZ test.
3. Then it clusters the active predictors using the precision matrix, which is described in Algorithm 1.
4. To finally control FDR, it proposes a novel importance score computation method for each cluster representative using Deep Neural Networks (DNNs) followed by a bootstrap resampling method.

I mainly have some questions and suggestions below:

1. This paper misses some key papers in the knockoff literature in their discussion. For example, the authors mentioned that "we adopt the cluster version of the FDR as the expected value of the "proportion of clusters that are falsely declared among all declared clusters". This is very related to the notion of group knockoff statistics for FDR control.  These main papers in the context of group knockoffs are not cited:

 a. A Prototype Knockoff Filter for Group Selection with FDR Control https://arxiv.org/pdf/1706.03400.pdf
 b. The knockoff filter for FDR control in group-sparse and multitask regression at https://proceedings.mlr.press/v48/daia16.html
 c. Deep-gKnock: Nonlinear group-feature selection with deep neural networks.

The authors should also briefly discuss the relationship of their paper to these aforementioned papers.

2. In general, the simulation study is well-designed and discussed. However, some other aspects should also be explored.

 a. In Appendix C.1, the equation of wo nonlinear relationship "Nonlinear additive" and "Nonlinear additive with interactions" should be emphasized and mentioned in the main text. The reason is single-index model is still similar to a linear relationship. The nonlinear additive and the one with interactions better support the results for a nonlinear relationship. The authors should emphasize that they have explored more complicated cases other than the single-index model.

b. Since the authors explore the features by clustering them and selecting cluster representatives, to test the effectiveness for this, the authors can design one simulation setting by varying within-group variation with rho ={0.2, 0.4, 0.6, 0.8} and between-group correlation rho={0.2, 0.4, 0.6, 0.8} to see if their cluster method is effective enough to differentiation the group membership of the features.
c. For the current existing simulation in Figure 4, instead of correlation rho=(0.1, 0.5, 0.9), more different rho values such as 0.2, 0.3, 0.4, 0.6, 0.7, 0.8 should also be explored to should the trend. Instead of using bar charts, it may be more effective to show performance comparison using multiple curves with different rho values with a fixed signal noise value like 5.

3. In terms of real data analysis:
 a.  Untangle the response mechanism of anticancer drugs using Knockoffs method has been explored in Zhao, Tingting, Guangyu Zhu, Harsh Vardhan Dubey, and Patrick Flaherty. "Identification of significant gene expression changes in multiple perturbation experiments using knockoffs." Briefings in Bioinformatics 24, no. 2 (2023). This work should also be cited.

 b. The authors describe the response variable as "For the response, we used the activity area (Barretina et al.,2012)", a little more information should be provided to describe what type of variable is activity area? Is it continuous or categorical, explain using one sentence about how genes could affect the activity area. Is this a classification task or a regression task?

c. In the last paragraph of page 13, the authors discuss some important genes selected by their method, is the corresponding coefficient positive or negative for these selected genes and how this is related with the upregulation or down-regulation of the genes and how this affects the response variable? More biological interpretations should be provided.  Also, is this consistent with existing literature?

Minor issue,
Under section 3.1, "we present .....", letter w should be capitalized.

---

### Author Response · Authors · 2023-07-28
**General response to Reviewers and Action Editors**

We express our heartfelt gratitude to the reviewers for their invaluable feedback and numerous constructive suggestions. Their rigorous evaluation and feedback have significantly strengthened the quality and contribution of our work.

In response to the reviewers' comments, we have made several improvements to the manuscript, which we would like to highlight:

**Notation and Definitions**: We have carefully revised the notation and definitions to enhance clarity and facilitate a better understanding of our proposed method.

**Algorithm Overview**: To aid readers in grasping the entire SciDNet workflow at a glance, we have added a schematic overview of the algorithm in the introduction section.

**Simulation Study**: We conducted an additional simulation study to showcase SciDNet's adaptability in addressing the uncertainty arising from high multicollinearity. This experiment highlights the stability of our approach across a wide range of autocorrelation coefficients.

**Real Data Analysis**: We have provided more comprehensive information about the datasets used in our real data analysis.

The manuscript now reflects a more comprehensive and refined presentation of our work, addressing the reviewers' comments effectively. We are excited to share our findings with the scientific community and believe that SciDNet will be a valuable addition to the existing literature on feature selection methods.

Thank you for your time and consideration. If you require further clarification on any specific aspect, please let us know.

Regards,

Authors

---

> ### Comment · Reviewer_j8Mo · 2023-08-11
> **Overall thoughts on paper: clarity and presentation could still be improved**
>
> Thank you for making a revision. I have commented on some specific issues related to my review in another comment, but wanted to respond here to have a discussion with all authors + reviewers. I think the biggest issues for this paper are:
>
> **Clarity:** both reviewer yi1K and I found the paper hard to follow. I don't think the method is all that complicated and it could probably be described in a much simple way. The authors made a few changes in their revision, but it appears these changes were very minimal. I think the paper could probably be made much better by being largely re-written.
>
> **Presentation:** beyond just the clarity of text, one thing I've noticed in my second look at the paper is that the figure quality could be improved a lot. For example:
>   - Figure 2 looks like a screenshot of an excel spreadsheet which has been resized so that the font aspect ratio is off (the text is unnaturally thin)
>   - All the figures have different styles (e.g. half seem to be from excel and half from matplotlib)
>   - Figure 9 has microscopic axis labels
>   - Figures are all raster images
>
> In machine learning, where authors are expected to do their own typesetting with LaTeX, I think the standard of good presentation is vector graphics whenever possible, no images of text, font sizes consistent with the main paper, and a consistent figure style. Even though this does not impact the underlying science, frankly I think the paper looks like something from a low-quality journal. Appearance matters, at least a little bit. I encourage the authors to improve this.
>
> **Overall:** I would like to know whether the other reviewers agree with what I wrote or have additional questions/comments for the authors.

---

> > ### Author Response · Authors · 2023-08-17
> >
> > Thank you for identifying the discrepancies in the plots. In the revised version of the manuscript, we have made enhancements to Figures 1, 2, and 8 for a better presentation. Furthermore, we have refined the axis labels in Figure 9.

---

### Decision · Action_Editors · 2023-09-29

**Recommendation:** Reject

**Comment:**

This paper introduces an innovative approach that integrates deep learning with screening and cleaning techniques to manage the false discovery rate (FDR) during feature selection for interpreting black-box models. The reviewers acknowledge the paper's relevance to TMLR's audience and recognize the potential utility of the proposed method. However, there are concerns about the clarity of the writing, which falls below TMLR's acceptance standards even after revisions. This significantly hinders the audience's ability to fully grasp the scientific contributions.

To enhance the paper's self-contained nature and assist readers, improvements are suggested in various aspects. The quality of figures could be refined, and greater clarity is needed in mathematical setups and claims, particularly regarding the notion of "minimal assumption." For example, the problem definition, expressed as "Our objective is to learn the sparsity structure by estimating S_0," lacks scientific clarity by not specifying the algorithm's outcome and evaluation criteria clearly.

Additionally, the incorporation and relevance of each assumption in designing the proposed method need elucidation. The lack of clarity in this regard makes it challenging to assess the necessity of each assumption. Since feature selection is a well-established field in the ML community, a more comprehensive literature survey and a comparative study are recommended to convince ML readers of the paper's distinctiveness.

**Audience:**

Yes.

**Claims And Evidence:**

Not completely. Clarity can be greatly improved.